



# Model-to-model data assimilation method for fine resolution ocean modelling

Georgy I. Shapiro[1], Jose Maria Gonzalez-Ondina[2]

[1] School of Biological and Marine Sciences, University of Plymouth, Plymouth, PL4 8AA, United Kingdom

[2] University of Plymouth Enterprise Ltd, Plymouth, PL4 8AA, United Kingdom

Correspondence to Georgy I. Shapiro ( gshapiro@plymouth.ac.uk)

## Abstract

An effective and computationally efficient method is presented for data assimilation in a high-resolution (child) ocean model, which is nested into a coarse-resolution good quality data assimilating (parent) model. The method named Data Assimilation with Stochastic-Deterministic Downscaling (SDDA) reduces bias and root mean square errors (RMSE) of the child model and does not allow the child model to drift away from reality. The basic idea is to assimilate data from the parent model instead of actual observations. In this way, the child model is physically aware of observations via the parent model. The method allows to avoid a complex process of assimilating the same observations which were already assimilated into the parent model. The method consists of two stages: (1) downscaling the parent model output onto the child model grid using Stochastic-Deterministic Downscaling, and (2) applying a simplified Kalman gain formula to each of the fine grid nodes. The method is illustrated in a synthetic case where the true solution is known, and the child model forecast (before data assimilation) is simulated by adding various types of errors. The SDDA method reduces the child model bias to the same level as in the parent model and reduces the RMSE typically by a factor of 2 to 5.

## Introduction

Fine resolution ocean modelling is becoming a ubiquitous practice to resolve important mesoscale and sub-mesoscale features such as eddies, fronts, boundary currents and localised upwellings which play important roles in ocean dynamics, see e.g. (GFDL, 2021; Dufour et al., 2015; Shapiro et al, 2010; T. Meunier et al, 2012.) Such localised models can be run by relatively small groups due to availability of good quality ocean models such as ROMS or NEMO to the wider oceanographic community (ROMS, 2021; NEMO, 2021). Local fine resolution models require initial and boundary conditions which can be obtained from good quality, but coarser resolution models run by major ocean modelling centres such as Mercator Ocean International (France) or Met Office (UK) via Copernicus Marine Service (CMEMS, 2021).

Due to inevitable approximations in the equations, numerical schemes, parameterisation and uncertainties in input data, ocean models tend to drift from reality. A process called data assimilation (hereafter DA) is often regarded as a way of keeping a model 'on the tracks' by constantly correcting it with fresh observations (DARC, 2021; Lorenc, 1986). DA is considered a cornerstone of all ocean analysis and forecasting efforts, where the rigorous and systematic combination of ocean

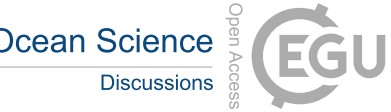

observations and ocean models yields an optimal estimate of the ocean state, see e.g. (Bell et al, 2000;
Moore et al, 2019). Numerical models of the ocean are able to assimilate oceanographic observations,
creating a dynamically consistent, complete and accurate description of atmosphere and ocean, see
e.g. (Ciavatta et al, 2018; Carrassi et al, 2018). Modern data assimilation is a complex process involving
statistics, methods of dynamical systems and numerical optimization, with an additional difficulty
which arises due to increasing sophistication of the environmental models. While the DA problem can
be formulated precisely, the solution is challenging because of the vast number of degrees of freedom
of the ocean state simulated by operational ocean models that represent many complex non-linear
processes (Moore et al, 2019; Dobricic and Pinardi, 2008; Ghil and Malanotte-Rizzoli, 1991). Thus,
despite data assimilation being nowadays ubiquitous in geosciences, it has so far remained a topic
mostly reserved to experts (Kubryakov et al, 2012; Carrassi et al, 2018).
The purpose of this paper is to develop a simple and computationally efficient method of DA which
can be implemented by smaller academic and operational centres which do not have resources of the
same scale as the major ocean forecasting institutions. The basic idea suggested in this study is to
assimilate data from a good quality but coarser resolution parent model instead of observations. The
parent model is assumed to be data-assimilating itself, ensuring that the observations, however
indirectly, will not allow the fine resolution (hereafter called child) model to deviate significantly from
the true state of the sea. In other words, the coarse model can be regarded as a physically aware
method for incorporating a sparse number of observations into a fine resolution regular grid.
While in principle a number of existing methods could be used for the model-to-model DA, this study
uses the Stochastic-Deterministic Downscaling (Shapiro et al, 2021) as its core element. The model-
to-model DA algorithm (hereafter called Data Assimilation with Stochastic-Deterministic Downscaling,
or SDDA) is described in the Data and Methods section. This section also describes the synthetic
idealised case where the true solution is known. Section Results uses the synthetic case for
demonstrating the performance of the SDDA method. The strengths and limitations of the SDDA
methods are discussed in the final section of the paper in comparison with the commonly used
combination of Hollingworth-Lönnberg and variational DA (VAR, also known as Optimal Interpolation)
methods (Hollingsworth and Lönnberg, 1986; Kalnay, 2003).

## Data and methods

This section presents the algorithm used for Data Assimilation with Stochastic-Deterministic
Downscaling and describes how the idealised cases are set and processed.

### The algorithm

The proposed algorithm is designed for assimilating data from an ocean model of coarser resolution
parent model, into a finer resolution child model.
A common approach to data assimilation into ocean (or atmospheric) model is based on minimising
the cost function $J$ given in Eq (1)

$$J(\boldsymbol{x}) = (\boldsymbol{x} - \boldsymbol{x}^b)^T \mathbf{B}^{-1}(\boldsymbol{x} - \boldsymbol{x}^b) + \left(\boldsymbol{y} - H(\boldsymbol{x})\right)^T \mathbf{R}^{-1}\left(\boldsymbol{y} - H(\boldsymbol{x})\right) \qquad (1)$$

where $\boldsymbol{x}$ is the (unknown) vector of best estimates of the true values, $\boldsymbol{x}^b$ is the model forecast before
data assimilation, $\boldsymbol{y}$ is the vector of observed values, H is the operator which projects data from the
model grid onto the locations of observations, $\mathbf{B}$ and $\mathbf{R}$ are error covariance matrices for the model
and observations respectively. In this paper we use the notation recommended in (Ide et al, 1997).



The value of $\boldsymbol{x} = \boldsymbol{x}^a$ which minimises the cost function $J$ is called the analysis, it is closest in an RMS
(root-mean-square) sense to the true state $\boldsymbol{x}^t$ (Bouttier and Courtier, 1999) This approach works well
for a relatively small (compared to model output) number of data. When the number of observations
is large, the matrices used in Eq (1) become very large and their inversion causes significant
computational problems.  For example, Bouttier and Courtier (1999) state that 'except in analysis
problems of very small dimension (like one-dimensional retrievals), it is impossible to compute exactly
the least-squares analysis.'
However, this study is related to assimilation of large amounts of data from one model to another.
For this purpose, an alternative cost function $J_S$ is proposed, namely
$$J_S(\boldsymbol{x}') = \left(\boldsymbol{x}' - \boldsymbol{x}'^b\right)^T \mathbf{B}^{-1}\left(\boldsymbol{x}' - \boldsymbol{x}'^b\right) + (S'(\boldsymbol{y}) - \boldsymbol{x}')^T \mathbf{R}^{-1}(S'(\boldsymbol{y}) - \boldsymbol{x}') \qquad (2)$$
where the primed variables denote deviations of the respective variable from their statistical mean,
and the statistical mean is designated by angle brackets
$$\boldsymbol{x}' \; = \; \boldsymbol{x} - <\boldsymbol{x}>$$
$$\boldsymbol{x}' \; = \; \boldsymbol{x}^b - <\boldsymbol{x}^b>$$
$$S'(\boldsymbol{y}) = S(\boldsymbol{y}) - <S(\boldsymbol{y})>$$

Here $\boldsymbol{y}$ is the vector of data from the parent model, and $S$ is the operator which projects (downscales)
the data from the parent model onto the fine grid of the child model. The best estimate for the true
value x is obtained by minimising the cost function
$$\nabla J_S = \mathbf{0}$$
Taking into account that matrices $\mathbf{B}$ and $\mathbf{R}$ are symmetric so that $\mathbf{B}^T = \mathbf{B}$ and $\mathbf{R}^T = \mathbf{R}$ one gets the
equation
$$\nabla J_S\left(\boldsymbol{x}'^a\right) = \left(\boldsymbol{x}'^a - \boldsymbol{x}'^b\right)^T (\mathbf{B}^{-1} + (\mathbf{B}^{-1})^T) - \left(S'(\boldsymbol{y}) - \boldsymbol{x}'^a\right)^T (\mathbf{R}^{-1} + (\mathbf{R}^{-1})^T) =$$
$$2\left(\boldsymbol{x}'^a - \boldsymbol{x}'^b\right)^T \mathbf{B}^{-1} - 2\left(S'(\boldsymbol{y}) - \boldsymbol{x}'^a\right)^T \mathbf{R}^{-1} = \mathbf{0} \qquad\qquad (3)$$

where $\boldsymbol{x}'^a$ is the fluctuation of the analysis vector around its statistical mean.
Taking the transpose of Eq (3) and dividing by two one gets

$$\mathbf{B}^{-1}\left(\boldsymbol{x}'^a - \boldsymbol{x}'^b\right) = \mathbf{R}^{-1}(S'(\boldsymbol{y}) - \boldsymbol{x}'^a) \qquad\qquad (4)$$
Let us introduce the error correlation matrices $\mathbf{C}_B$ and $\mathbf{C}_R$ for the child and downscaled parent
models:
$$\mathbf{B} = \mathbf{V}_B \mathbf{C}_B, \qquad \mathbf{R} = \mathbf{V}_R \mathbf{C}_R$$


where $\mathbf{V}_B$, $\mathbf{V}_R$ are the diagonal matrices containing respective error variances at each fine grid
node. The inverted diagonal matrices are also diagonal. Equation (4) then becomes

$$\mathbf{C}_B^{-1}\mathbf{V}_B^{-1}(\boldsymbol{x}'^a - \boldsymbol{x}'^b) = \mathbf{C}_R^{-1}\mathbf{V}_R^{-1}(S'(\boldsymbol{y}) - \boldsymbol{x}'^a) \qquad (5)$$

Correlation matrices both for the child and downscaled parent models relate to the same resolution
and the same area of the ocean, hence it is reasonable to use the same correlation function, e.g. a
Gaussian of a certain length scale and therefore the same correlation matrix for both models, i.e.
$\mathbf{C}_B = \mathbf{C}_R = \mathbf{C}$. After pre-multiplying Eq (5) by $\mathbf{C}$ and re-arranging the terms, one gets

$$(\mathbf{V}_B^{-1} + \mathbf{V}_R^{-1})\boldsymbol{x}'^a = \mathbf{V}_B^{-1}\boldsymbol{x}'^b + \mathbf{V}_R^{-1}S'(\boldsymbol{y}) \qquad (6)$$

or   $$\boldsymbol{x}'^a = (\mathbf{V}_B^{-1} + \mathbf{V}_R^{-1})^{-1}(\mathbf{V}_B^{-1}\boldsymbol{x}'^b + \mathbf{V}_R^{-1}S'(\boldsymbol{y})) \qquad (7)$$

Using the commutative properties of diagonal matrices and the following identity

$$(\mathbf{V}_B^{-1} + \mathbf{V}_R^{-1})^{-1} = \mathbf{V}_R(\mathbf{V}_B + \mathbf{V}_R)^{-1}\mathbf{V}_B, \qquad (8)$$

the solution for the analysis state given by Eq (7) can be re-written as
$$\boldsymbol{x}'^a = (\mathbf{V}_B^{-1} + \mathbf{V}_R^{-1})^{-1}(\mathbf{V}_B^{-1}\boldsymbol{x}'^b + \mathbf{V}_R^{-1}S'(\boldsymbol{y})) = \mathbf{V}_R(\mathbf{V}_B + \mathbf{V}_R)^{-1}\boldsymbol{x}'^b +$$
$$\mathbf{V}_B(\mathbf{V}_B + \mathbf{V}_R)^{-1}S'(\boldsymbol{y}) \qquad (9)$$

Eq (9) can be interpreted as a zero-dimensional Kalman gain formula applied to the fluctuations of
state variables in the parent and child models at each fine grid node independently.  The term 'zero-
dimensional' reflects the fact that in this case, the matrices used in the Kalman gain formula have the
size of 1x1, i.e. are reduced to a scalar. For a single $\boldsymbol{x}'_i$ element of the state vector $\boldsymbol{x}'$ Eq (9) gives
$$x_i'^a = \frac{V_{Rii}}{V_{Rii}+V_{Bii}} x_i'^b + \frac{V_{Bii}}{V_{Rii}+V_{Bii}} S'(\boldsymbol{y})_i \qquad (10)$$
For the $S'$ operator, which downscales the fluctuations of the field variable from the coarse to fine
grid, we use the Stochastic-Deterministic Downscaling (SDD) method developed in (Shapiro et al,
2021). This method calculates the weighting coefficients required for downscaling from coarse to fine
grid by adopting the original algorithm of optimal interpolation (Gandin, 1959, Gandin 1965) and
statistical properties of field variable estimated from the parent model outputs.  As a result, it
generates lower errors during downscaling than if using interpolators with prescribed coefficients





such as linear or polynomial. The spatial correlation matrix $\mathbf{C}$ is not explicitly included in expressions
(9) and (10), but it is used at the downscaling stage of data assimilation, hence the covariances
between data at the parent model grid are implicitly present in the operator $S'$. The ensemble
statistical means at each grid node which are required by the SDD method are estimated using the
ergodic hypothesis (Stull, 1998), i.e. by spatial averaging over a small trial area around the node at
the same time point, see (Shapiro et al 2021) for details.
Equation (10) performs data assimilation on the fluctuations, however, the statistical means for the
parent and child model are, in a general case, different. There are at least two options of how to
obtain the mean for the analysis. Option one is to apply to the means the same Kalman filter as for
fluctuations. The problem here is that variances of statistical means are unknown, and hence the
weighting coefficients in Eq (9) or Eq (10). Option two is to assume that the parent model is of good
quality, and it has been debiased as much as possible during its own data assimilation cycle. Therefore,
it is reasonable to replace the (potentially biased) statistical mean from the child model with the
statistical mean of the parent model.
This gives the final equation for calculation of the analysis state
$$\boldsymbol{x}^a = \mathbf{V}_R(\mathbf{V}_B + \mathbf{V}_R)^{-1}\boldsymbol{x}'^b + \mathbf{V}_B(\mathbf{V}_B + \mathbf{V}_R)^{-1}S'(\boldsymbol{y}) + <S(\boldsymbol{y})> \qquad (11)$$
Or for an element of the state vector at a particular fine grid node $i$ ($i = 1 \dots N$) where $N$ is the
number of grid nodes in the child model one gets
$$x_i^a = \frac{V_{Rii}}{V_{Rii}+V_{Bii}}\boldsymbol{x}'^b{}_i + \frac{V_{Bii}}{V_{Rii}+V_{Bii}}S'(y)_i + <S(y)>_i \qquad (12)$$
To summarise, the SDDA model-to-model data assimilation procedure includes two steps. Firstly, to
downscale the parent model output from the coarse grid onto the child model fine grid using the SDD
method. The result is that at each fine grid node there are two values of the same state variable. The
best estimate of the true field is obtained at the second step by combining these two values using a
zero-dimensional Kalman filter. This algorithm is computationally efficient as it does not require
inversion of large matrices or solving a very large system of algebraic equations at the second step,
something that is required if using the variational methods based on Eq (1). The inversion of
correlation matrices to obtain the weight coefficients for the SDD step can be done only once at the
beginning of the model run, as these coefficients do not depend on time. Another benefit of the
described method is that the correlation matrices for the downscaling of the parent model have a
relatively small rank and condition number, and their inverse counterparts can be calculated without
the need of any type of matrix regularisation, just using double precision for the computer
representation of the variables. This is due to the fact that the SDD method assumes local
homogeneity and isotropy of statistical properties of the field variable and that the correlation
function is set to zero for distances larger than a threshold value. More details on the philosophy and
the algorithm of the SDD method can be found in (Shapiro et al, 2021).
Synthetic Idealised case
In this section the SDDA algorithm is illustrated in a synthetic idealised case where the true solution is
known. The task is to generate an analysis state using the fine-resolution (child) model forecast and
the output from a good quality data assimilating parent coarse model which will be used instead of
actual observations. It should be noted that, even if both coarse and fine model were perfect, there
would be some unavoidable discrepancies, or *representativeness errors*, between the two models due
to different meshes they use (Bouttier and P. Courtier, 1999).



The parent model output is simulated by sampling the true solution (a prescribed function) on the
parent model coarse grid, to which random noise might be added. The fine-resolution forecast is
simulated by sampling the same true solution on the fine grid, adding random noise, bias, and some
spatial shift. The latter is to simulate the 'double penalty effect' which is common to fine resolution
models, see e.g. (Zingerlea and Nurmib, 2008).
Three distinct examples are considered, all of them relate to data assimilation of one field variable at
a single computational surface, whether it be a geopotential ('horizontal') level, such as used in Bell
et al. (2000), or a curved level such as used in sigma (Mellor and Blumberg, 1985) or multi-envelope
vertical coordinate system (Bruciaferri et al, 2018). The properties of the SDDA method are analysed
for: (i) an ocean front, (ii) an isolated isotropic eddy, (iii) a system of densely packed anisotropic
mesoscale eddies. In all examples the parent model has resolution of $\Delta x_p = \Delta y_p = 10$ km, and the
child model has resolution of $\Delta x_p = \Delta y_p = 10$  km. For all the examples, the correlation function
which is used to calculate the covariance matrix $\boldsymbol{C}$ was set to zero for distances greater than twice the
correlation length of $L = 17$ km, and the trial area used to calculate local spatial averages was a square
of 68 x 68 km$^2$ centred at each node.


a) Ocean front

An ocean front is a narrow area separating two water masses, and it has significant impact on
horizontal and vertical exchanges, see e.g. (Fedorov,1986).  This example is set in a square domain of
200x200km, and the front extends in the meridional ($y$) direction at the centre of the domain.
The true solution is set to be in the form
$$F = A \tanh\left(\frac{x}{L_x}\right) \tag{13}$$
Where $A$ is the amplitude of the front and $L_x$ is its half-width. For this exercise $= 1$ , and $L_x$  ranges
between 6 and 40 km.
b) Isolated mesoscale eddy
Mesoscale eddies are a ubiquitous feature of the World Ocean. Originally, they were thought to exist
only next to jet currents such as the Gulf Stream, however since the 1960s-1970s it became clear that
mesoscale eddies exist nearly anywhere in the ocean.  Most of the kinetic energy of the ocean is
contained in mesoscale eddies (Robinson, 1983). In this example an isotropic eddy is placed in the
centre of a domain of 200×200 km and the true solution is set to be in the form

$$F = A \exp\left\{-\left[\left(\frac{x}{L_e}\right)^2 + \left(\frac{y}{L_e}\right)^2\right]\right\} \tag{14}$$
Where $A$ is the amplitude of the eddy and $L_e$ is its radius. For this exercise $A = 1$, and $L_e$ ranges
between 6 and 46 km.

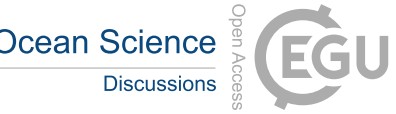
c)  Multiple mesoscale eddies
For this example, let us consider a square domain 1000×1000 km and take the true solution for a
variable $F$ in the form of multiple anisotropic mesoscale eddies

$$F = \sin\left(\frac{\pi x}{L_x}\right) \sin\left(\frac{\pi y}{L_y}\right) \tag{15}$$
where $L_x$ is the eddy 'radius' in the $x$-direction, and $L_y = 105$ km is the eddy 'radius' size in the $y$-
direction. Let us consider the range of eddy sizes $L_x$ between 12 and 24 km. At $L_x$ =24 km the parent
model can be classed as eddy-permitting as it has 2.4 grid point over the smaller 'diameter' of the
eddy. At $L_x$ =12 km the parent model is not even eddy-permitting but only showing an 'embryonic'
representation of the eddy. The child fine-resolution model is eddy resolving for any $L_x$ within the
chosen range of eddy sizes.

Results
The SDDA data assimilation method detailed in the section Algorithm above was applied to the
simulated child model forecast in order to create analysis for the next forecasting cycle. The results
for four examples are presented in this section: an ocean front, an isolated single eddy, a set of
multiple eddies.
A)  Ocean front
This example uses Eq (13) for the representation of an ocean front in the meridional direction. Fig.1
shows a map of the true solution $F$ for a sharp ocean front at $L_x$=6 km (see Fig.1(a)) , its representation
by the parent model ( Fig.1(b)) , by child model forecast  before data assimilation (Fig. 1(c)) as well as
transects across the front. The following parameters of added errors are used to simulate the child
model forecast before data assimilation: normally distributed noise with standard deviation 0.15,
positive bias of 0.3, and shift of the field by 4 km to the west.

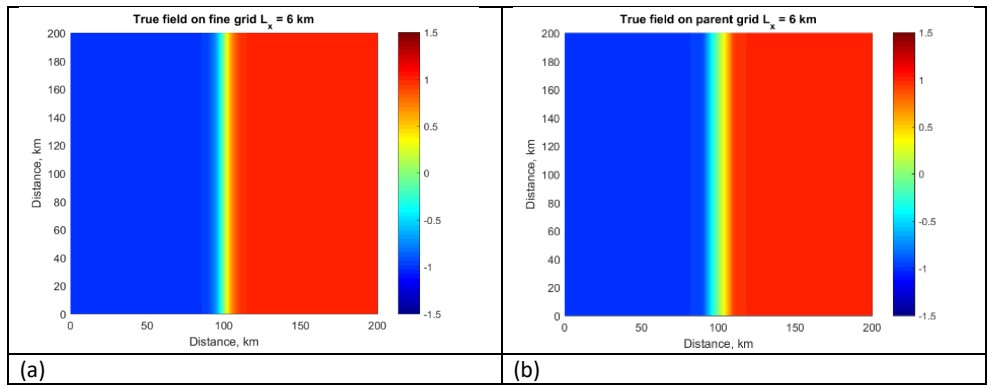

(a)                                                    (b)



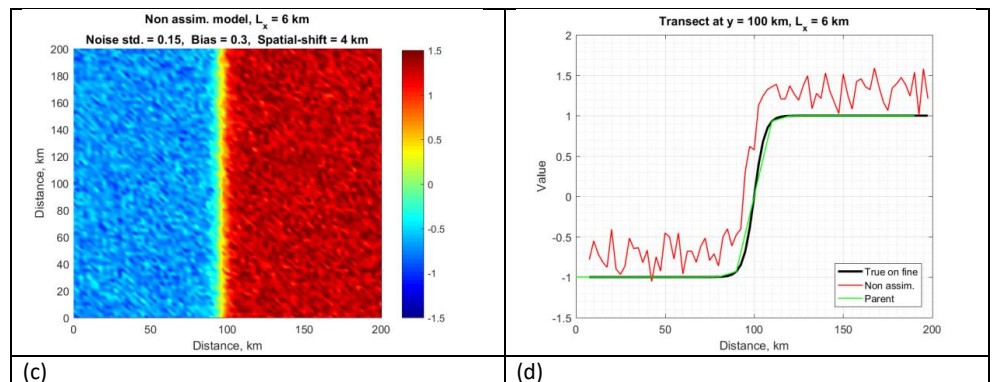

(c) (d)

**Fig. 1:** *Ocean front of width $L_x = 6$ km. (a) True field on the fine mesh, (b) parent model output, (c) simulated child model before data assimilation, (d) transects of the previous fields along the line at $y = 100$ km.*

Even at a resolution of 10 km that does not resolve the structure of the front (half-width of 6 km) the parent model gives a reasonable representation of areas outside the front where the changes in the state variable are smooth- see Fig. 1(b). However, as expected, the width of the front is exaggerated due to insufficient resolution. The child model forecast shown in Fig. 1(c) is noisy and clearly shifted westward relative to the true state.

Fig.2 shows the results of assimilation of output from the parent model using the SDDA method.

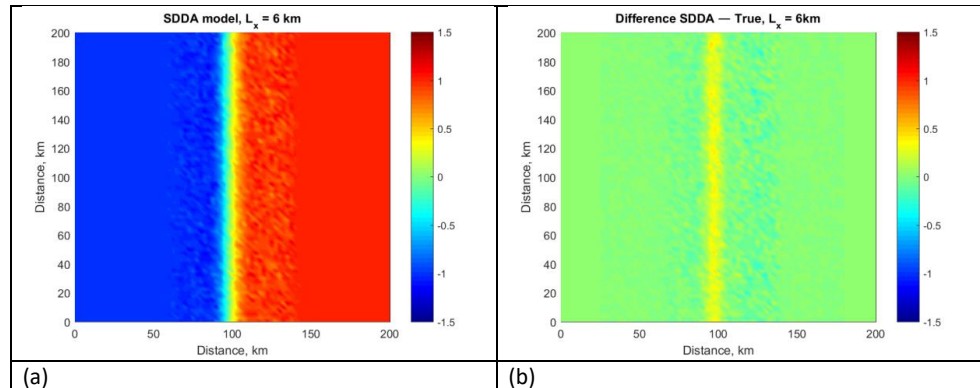

(a) (b)





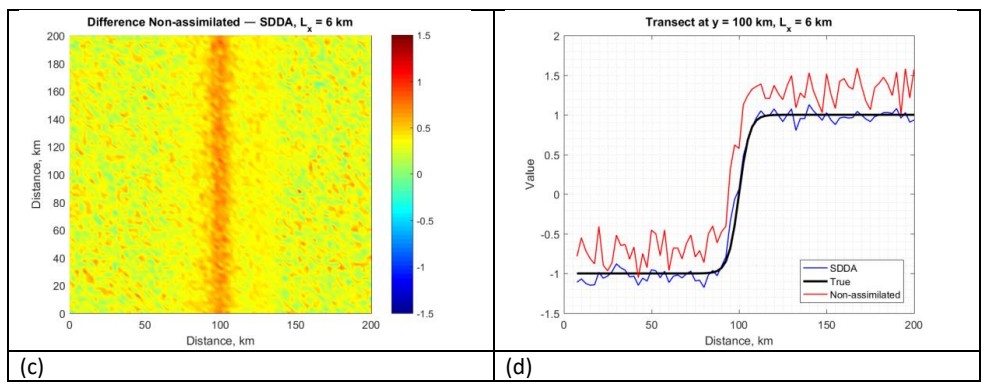

(c)  (d)

263

***Fig. 2.** Assimilation results for the ocean front shown in **Figure 1:** (a) child model after SDDA*
*assimilation, (b) difference between the assimilated model and the true field, (c) difference between*
*the non-assimilated and assimilated child models, (d) zonal transects of the previous fields along a*
*line at $y = 100\ km$.*

The analysis state (after assimilation) removes the spatial shift and bias and reduces the noise even
for such a sharp front where the resolution of the parent model is inadequate. For comparison, Fig.3
shows the improvement provided by the SDDA assimilation for fronts of different sharpness, with $L_x$
= 10, 20, 30 and 40 km.  In all examples the errors in non-assimilating child model forecasts are
simulated by adding bias = 0.3, random normally distributed noise with standard deviation (STD) =
0.15 and spatial shift of 4 km to the west.













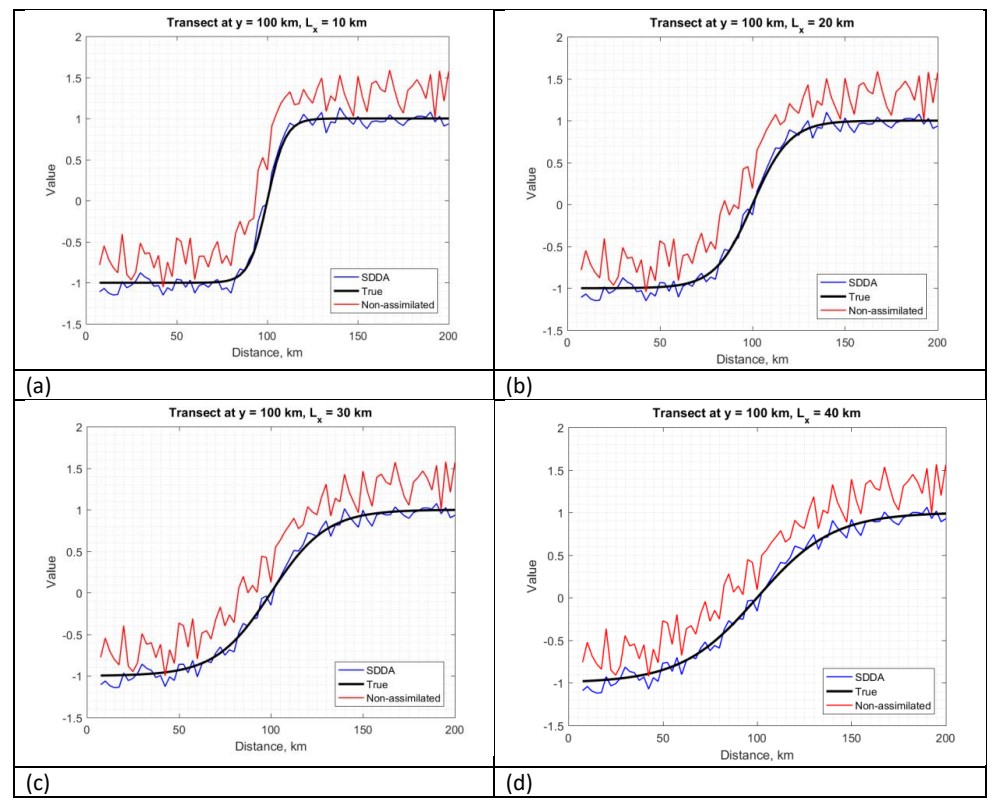

**Fig.3.** *Results of SDDA assimilation for fronts of different sharpness: (a) $L_x = 10$ km, (b) $L_x = 20$ km, (c) $L_x = 30$ km, (d) $L_x = 40$ km. The curves present data for the true field (black), non-assimilated noisy child model (red) and child model after SDDA assimilation (blue).*

Comparison of panels (a)-(d) in Fig. 3 and panel (d) in Fig. 2 shows that the improvement due to data assimilation from the parent model is achieved both for sharp fronts not resolved by the parent model and for smooth fronts.

Fig.4 shows how the bias and RMSE of the child model against the true state change with the sharpness of the front before and after data assimilation.

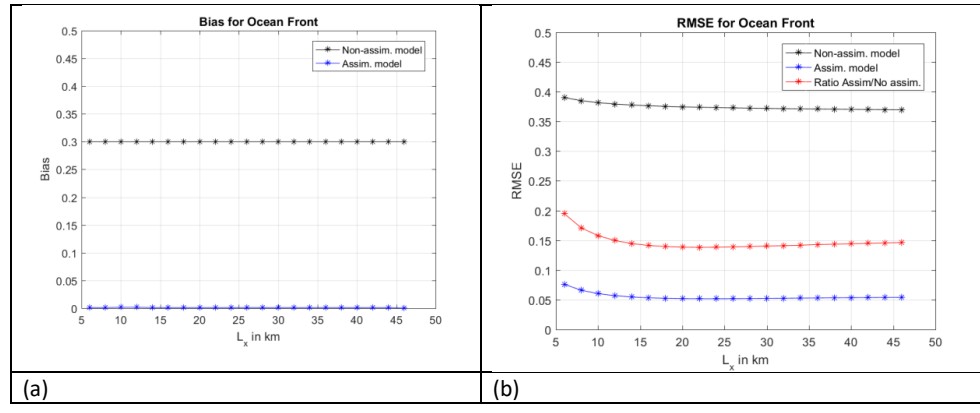





**Fig. 4.** *Plots of bias (a) and RMSE (b) for the Ocean Front case with different front sharpness for non-*
*assimilated (black) and assimilated (blue) child models.*

After applying the SDDA method, the bias is practically removed with the remaining values being of
the order 5 10$^{-3}$ or less. The RMSE calculated against the true state is more than four times lower than
before data assimilation as shown by the red line in Fig.4(b).
Similar properties are demonstrated in the example of a single eddy.
B) Single eddy
Fig.5 shows a map of the true solution *F* for an axisymmetric mesoscale eddy with the radius of $L_e =$
16 km, its representation by the parent coarse model and by the child model before data assimilation.
It also shows transects across the eddy centre. The parameters of added errors are the same as in the
previous example: normally distributed noise with a standard deviation of 0.15, positive bias of 0.3,
and shift of the field by 4 km to the west.







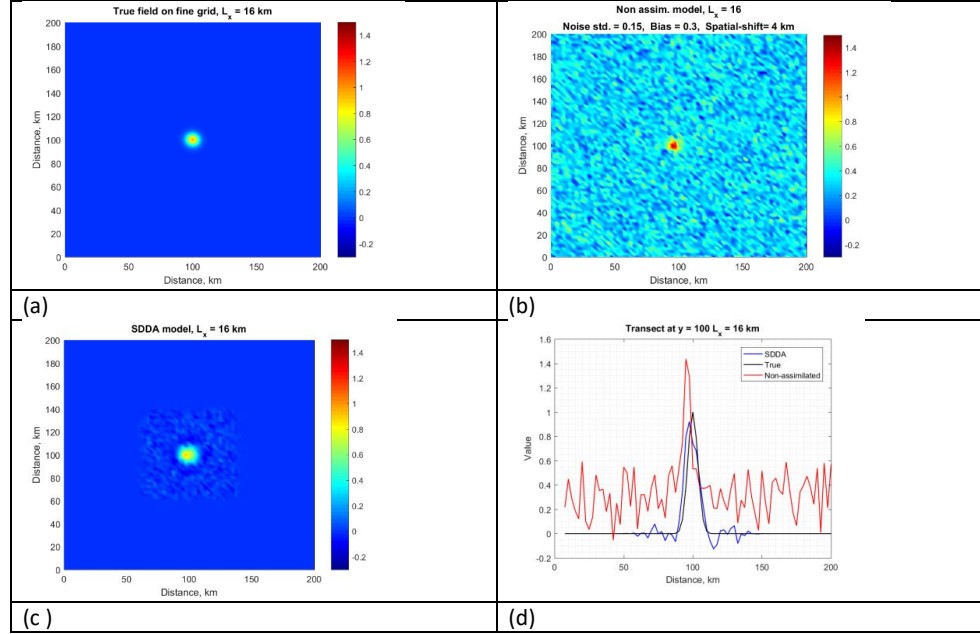



**Fig.5.** *Results of SDDA assimilation for a single isolated eddy: (a) true field sampled on the fine grid, (b) child model before data assimilation, (c) child model after SDDA data assimilation, (d) transects showing the same fields as in (a-c).*

The SDDA method reduces errors nearly to zero outside of the eddy and greatly reduces them inside the eddy (Fig 5 (b,d)). The bias is eliminated from 0.3 before data assimilation to 0.002 or less after assimilation. The reduction of RMSE for various values of eddy radius in the range of 6 to 46 km is shown in Fig 6.

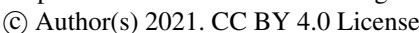

**Fig. 6** *Dependence of RMSE for different eddy sizes: before data assimilation (black), afte SDDA assimilation (blue), the ratio of RMSE after and before data assimilation (red)*

The next example illustrates the qualities of the SDDA method for a domain of 1000×1000 km packed with anisotropic eddies.

C) Multiple eddies

Fig.7 shows maps of the field variable F from the parent and child models with the following parameters of added errors to simulate child model forecast before data assimilation: normally distributed noise with a standard deviation of 0.15, positive bias of 0.3, and spatial shift of the field by 4 km to the west.

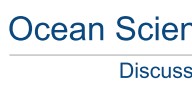 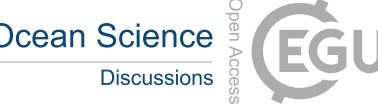

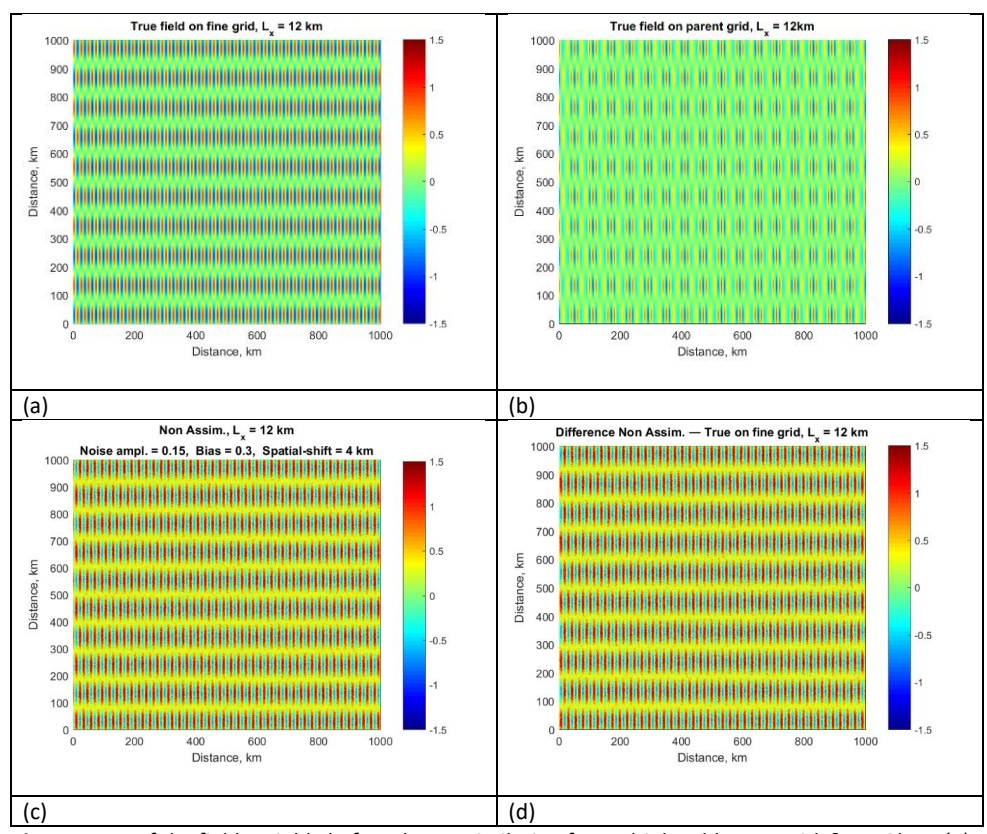

**Fig. 7.** *Maps of the field variable before data assimilation for multiple eddy case with $L_x$ =12km: (a) True field on fine grid, (b) True field on parent grid, (c) Non-assimilated model, (d) Difference between non-assimilated model and true field.*

Fig.7(b) shows that the parent model generally underestimates the true field shown in Fig.7(a) due to representativity errors. The difference between the non-assimilated child model forecast and true state shown in Fig. 7(d) is substantial showing the RMSE =0.61 which is about 30% of the range of the true field. The anatomy of the differences between the true state, and outputs from parent (coarse) and non-assimilated child models are shown on a zoomed-in section of the zonal transect in Fig.8. The largest errors produced by non-assimilated model are due to spatial shift and bias. The errors produced by the parent model are exclusively due its insufficient resolution as we assume that otherwise the parent model is perfect.

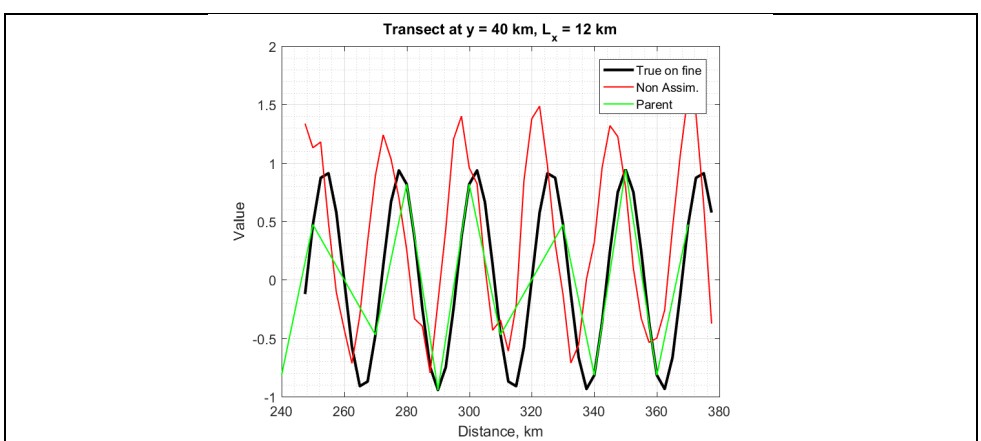

**Fig. 8.** *Transects of true field on the fine grid (black) and outputs from the parent (green) and non-assimilated child (red) models*

The SDDA method presented in the previous section is then applied to assimilate the data from the parent model into child model forecast.

Fig.9 shows the results of data assimilation (analysis) and a map of differences between the assimilated child model and true field.

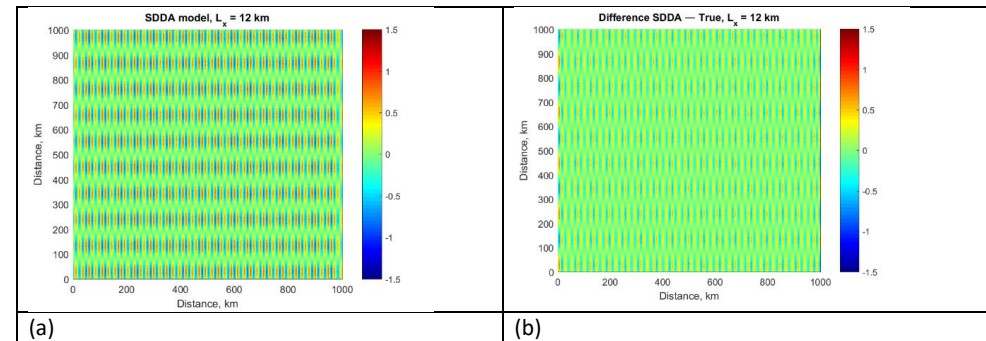

(a)                                                                                          (b)

**Fig.9.** *Maps for the multiple eddies case with width $L_x = 12$ km: (a) SDDA assimilated model, (b) Difference between SDDA assimilated model and true field. Parameters of the non-assimilated model are the same as in Fig.7 and Fig.8.*

The details of the improvement achieved by SDDA data assimilation are shown on a zoomed-in zonal transect in Fig. 10**.**


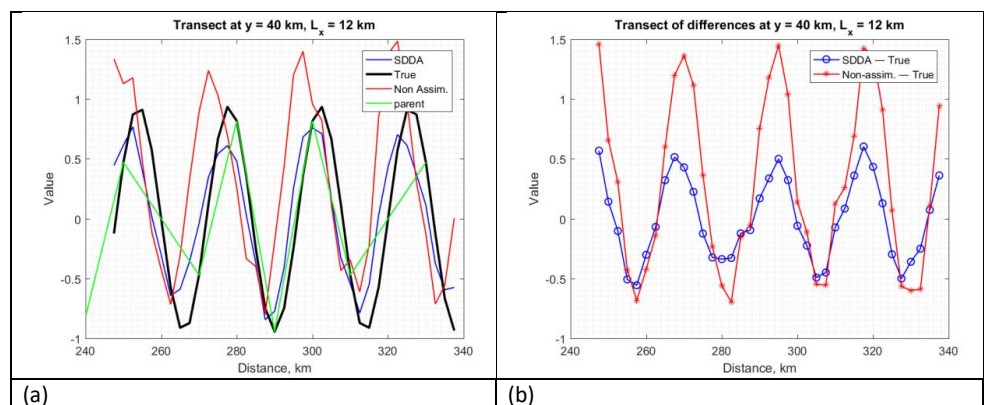

(a)                                              (b)

**Fig.10.** *Transects at $y = 40$ km for the multiple eddies case with $L_x = 12$ km. (a) Transects of SDDA*
*model output (blue), true solution (black), parent model (green) and the non-assimilated child*
*model(red); (b) differences between assimilated (blue) and non-assimilated (red) models and the true*
*solution.*

Data assimilation partially reduces the spatial shift which simulated the double penalty effect common
to fine-resolution models. The reduction of spatial shift is due to the properties of the SDD component
of the SDDA, see (Shapiro et al, 2021) for details. The significant reduction of the bias is due to the
assimilation part of the SDDA algorithm, however it also causes some reduction in the amplitudes of
the eddies – see Fig. 10(a).  The remaining errors shown in Fig. 10(b) are mainly due to incompletely
corrected spatial shift and random noise in the non-assimilated model. The RMSE of the assimilated
model is lower at 0.25 vs. a non-assimilated value of 0.61, and the bias is reduced by orders of
magnitude from 0.3 to - 1.3 $10^{-5}$.
Fig.11 shows the results of the sensitivity analysis with different sizes of eddies and levels of various
sources of errors in the non-assimilated child model forecast- noise, bias, and shift.

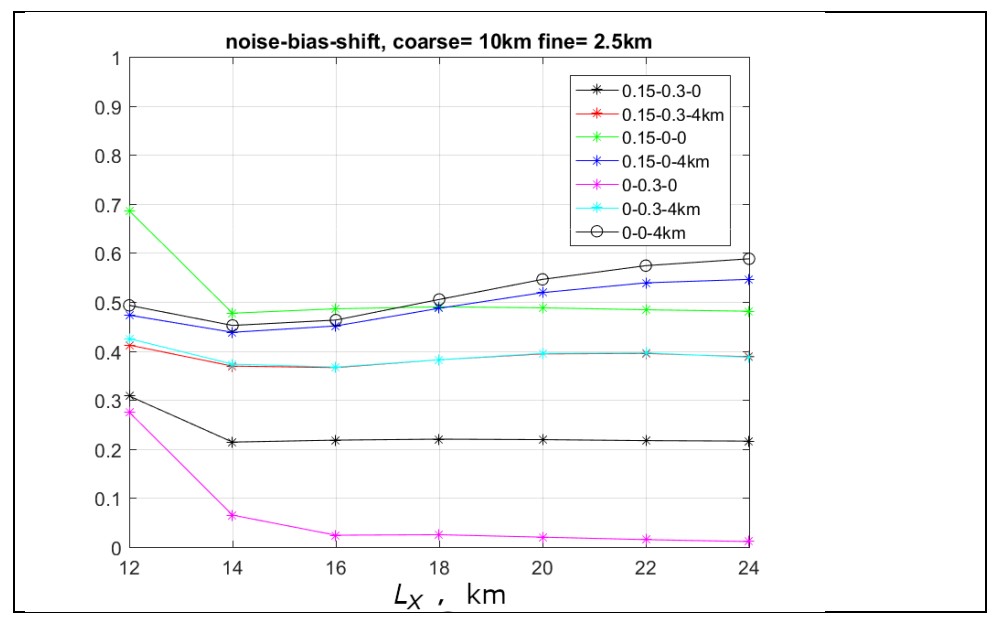

***Fig. 11** Ratio of RSME values of the child model relative to the true solution after and before SDDA*
*data assimilation as a function of eddy radius in the zonal direction $x$. Ratios are calculated for*
*different combinations of errors generated by the child model – random noise (STD equal to either*
*zero or 0.15), spatial shift (zero or 4 km), and bias (zero or 0.3). Percentages are shown with respect*
*to the amplitude of the true signal. The first number in the legend shows the level of noise, the*
*second shows the bias, and the third one shoes the spatial shift.*

In this example, the SDDA method works best when the only source of error is the bias which is
removed nearly completely. The second-best results are achieved when only random noise and bias
are present but not the spatial shift. The RMSE ratio for cases containing errors due to spatial shift
grows slightly at larger eddy sizes. This is due to the fact that a relatively small shift of 4 km does not
distort large eddies to the same extent as small eddies even in the non-assimilated model. Generally,
the curves in Fig.11 show that after application of SDDA data assimilation algorithm the RMSE is
reduced by half or better. Bias between the child and parent models is reduced by orders of
magnitude. Assuming that the parent data assimilating model is unbiased, it means that the fine
resolution model becomes unbiased after application of SDDA data assimilation process.

D)  Effect of errors in the parent model
The next example shows the quality of the SDDA method when the parent model is still noisy even
after its own data assimilation cycle. To investigate how such noise impacts on the reduction of RMSE
in the child model when the SDDA data assimilation is applied, a quite large random noise was added
to the parent model output with a standard deviation of 0.1, i.e. 10% of the signal. The results for the
ocean front and multiple eddies examples are shown in Fig.12 and Fig.13.



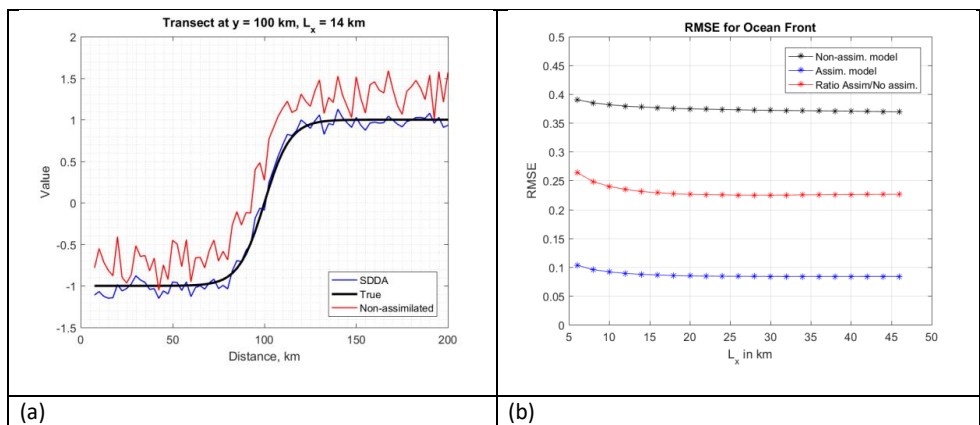


**Fig. 12.** *Ocean front case of $L_x = 14$ km, with noisy parent model (added noise STD = 0.1) and noisy*
*non-assimilated child model (added noise STD = 0.15, bias = 0.3 and shift of 4km to the west). (a)*
*Transects at $y = 100$ km of assimilated (blue), non-assimilated (red) child models and the true field*
*(black); (b) RMSE calculated for non-assimilated (black) and assimilated (blue) child models and the*
*ratio between them at different front stepnesses.*

The transect in Fig.12 shows that after SDDA, the front is well represented with small random noise
even when the parent model is quite noisy. The improvement is five to ten-fold and is consistent
across the range of front widths from 6 to 46 km.

403

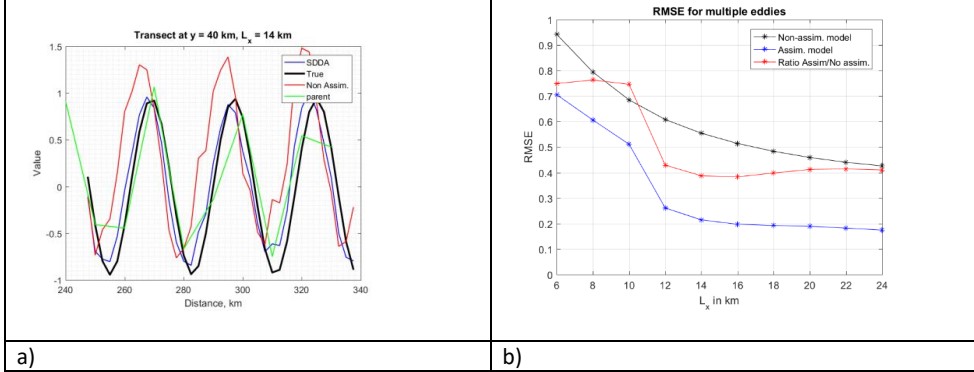

**Fig.13.** *Multiple eddy case of $L_x = 14$ km with noisy parent model (added noise STD= 0.1) and noisy*
*non-assimilated child model (added noise STD = 0.15, bias = 0.3 and shift of 4 km to the west). (a)*
*Transects at $y = 100$ km of true field (black), parent (green), assimilated (blue) and non-assimilated*
*(red) child models, (b) RMSE for different eddy sizes and the ratio between them.*

408

In case of multiple eddies, the improvement is less dramatic, however for eddy diameters larger than
12 km the RMSE is improved approximately by half. For eddy sizes 10 km and less the improvement
due to assimilation of the parent model is limited, which is expected as such eddies are strongly
distorted by the parent model of only 10 km resolution.





## Discussion

The presented examples show the data assimilation cycle using the SDDA method when the data used
for assimilation are not coming from observation as in the common assimilation methods but from a
coarser parent model. The examples relate to a synthetic situation when the true values of the state
variable are known. The forecast by fine-resolution child model is simulated by sampling the true field
on the model's fine grid and adding various sources of errors-random noise, bias, and spatial shift. The
output from the coarser parent model is regarded as computer generated 'observations' and is
simulated by sampling the true field on the coarser grid to which random noise can be added. In
practice, the parent model is assumed to be a good quality, dynamic ocean model assimilating
observational data. Therefore, the actual field measurements are used in the SDDA method indirectly
– instead of assimilating observation, the child model assimilates data from the coarse model, which
in turn assimilates data from observations.
The data from the parent model is assimilated by a two-step SDDA process into the forecast produced
by the child model to produce the analysis state which is then used as initial condition for the next
forecasting cycle. The first step of the SDDA method is the application of Stochastic-deterministic
downscaling presented in (Shapiro et al, 2021). As a result, at each fine grid point there are two
generally different values of the field variable. The second step is to combine these values separately
at each grid point using a suitable data assimilation method. A comprehensive overview of data
assimilation methods is given in the paper by Carrassi et al (2018). In this paper, a zero-dimensional
Kalman filter is used, similar to what have been done in atmospheric chemistry (Adhikary et al, 2008).
For the full 2D or 3D field this is equivalent to have a strictly diagonal background error covariance
and 'observation' error covariance matrices.
The gain formula used in Kalman filter requires the knowledge of error variances. There are a number
of approaches to estimate the variances in an ocean model, e.g. the Canadian, the NMC and the H-L
methods. These methods do not separate completely the variances of child model and observations
due to model errors and the natural variability of the ocean state among other reasons. In this paper
the variances are assessed using a widely used ergodic hypothesis, see e.g. (Stull, 2003), and the
ensemble statistical mean at each grid node is estimated by spatial averaging in a small trial area
around the node at the same time point. Such scheme is fast and does not consume significant
computational resources. If necessary, it can be extended by including data from preceding time
points, however these data may not be statistically independent and hence the potential
improvement requires further study.
The mechanism of improvement of fine model outputs by the SDDA method can be seen from the
analysis of spectral characteristics of the downscaled parent model, fine model forecast (before data
assimilation) and analysis (after data assimilation) in Fig.14. The bias represented by the peak at zero
wavenumber is removed at the data assimilation step of SDDA. Noise in the range below Nyquist
wavenumber is reduced by melding the noisy child model forecast with the clean parent model data.
It should be noted that the SDD downscaling honours the data of the parent model on the child nodes
that coincide with a coarse grid node. A small peak between the coarse and fine grid Nyquist
wavenumbers is an artefact created at the SDD step, see (Shapiro et al, 2021) as seen from the
spectrum of the downscaled field. Other than this peak, the downscaled field has much lower level of
noise than the child model forecast. Reduction of noise at wavenumbers higher than the parent model
Nyquist frequency is due to the low level of noise generated at the downscaling step of the process.





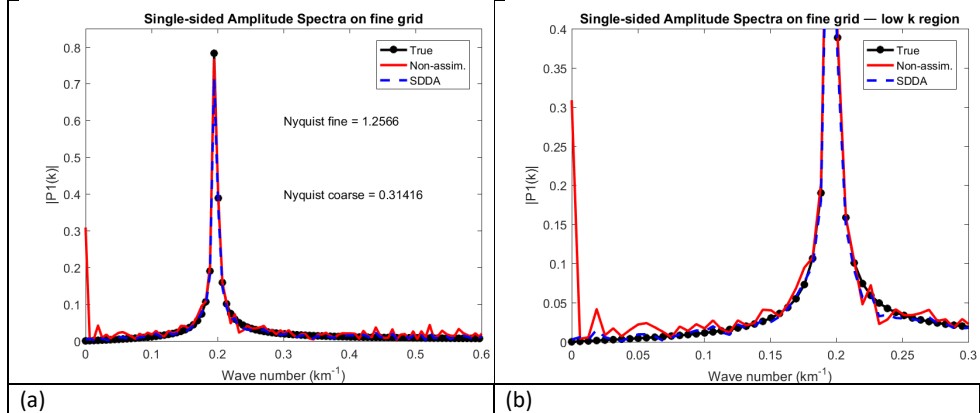

(a) (b)


**Fig.14.** *Amplitude spectra on a zonal transect at y=40 km for the example of multiple eddies of $L_x =$*
*16 km. The STD of noise in the non-assimilated child model is 0.15, bias is 0.3 and the spatial shift is 4*
*km to the west. The spectra are calculated for the true field, the child model forecast (before*
*assimilation) and analysis (after assimilation). (a) Spectra in the full range of wave numbers, (b)*
*zoomed in area for low wave numbers.*

The qualities of the SDDA method can be illustrated by comparison with existing data assimilation
techniques applied to mode-to-model data assimilation. Let us consider the combination of H-L
(Hollingsworth and Lonnberg, 1986) and variational methods, which is considered to be the most
commonly used estimation technique, see e.g. (Stewart et al, 2014; Carrassi et al 2018). There are a
number of variants of this method, here we applied the 'practical' algorithm described in the textbook
by Kalnay (2003), which will be named hereafter as the 'standard' method. The 'standard' method
consists of the following stages: calculate innovations (differences between observations and model)
at each observational point, estimate the covariances between innovations at different locations, fit
the best-fit curve (usually Gaussian) using all covariances except at zero distance, estimate the model
and observation error variances using the value where the best-fit curve intersects the r=0 vertical line
on the covariance plot. In the practical implementations of this algorithm the model and observation
error variances are assumed spatially homogenous (Kalnay, 2003). The model variances and the best-
fit curve are used to create the background error covariance **B**-matrix, and the diagonal observational
error covariance **R**-matrix. Due to spatial homogeneity, all diagonal elements in B and R matrices are
the same but different between matrices. The H-L method requires combination of innovations into
spatial bins, so that all innovations within a certain bin are allocated to the same distance from the
central point. In practice the covariances are calculated by averaging individual products of
innovations over a period of time instead of statistical averaging assuming the ergodic hypothesis
(Stull, 2003). In the model-to-model DA approach the role of observations is played by the output
from the parent data assimilating model.
For comparison between the SDDA and 'standard' methods we have selected the 'multiple eddies
example' (as in section Results, subsection C). The covariances required to build the **B** and **R** matrices





were estimated as follows. We generated 150 random realisations of the child model outputs which
were the sum of for the true field given by Eq (15) with $L_x = 40$ km and $L_y = 105$ km in a subdomain
of 300×300 km, and the following errors: (i) random noise with standard deviation of 0.15; (ii) random
spatial shift with a standard deviation of 4 km, (iii) constant positive bias of 0.3. Random components
were normally distributed with zero mean. Due to significantly larger computing resources required
by the 'standard' method compared to SDDA, the 'standard'  DA was done in the reduced domain
300×300 km instead of 1000×1000 km for the SDDA method. The bins required for the H-L method
were 1 km in size.
The innovations were computed as $\boldsymbol{d} = \mathbf{H}\boldsymbol{x^b} - \boldsymbol{y}$,  where $\mathbf{H}$ is a subsampling matrix operator
composed only of ones and zeroes. Following the 'standard' method we used a simple isotropic
Gaussian function for the covariance of innovations at every parent model grid point $x$:

$$C_x(r) = a \, exp(-r^2/D^2)$$   (16)
where $a$ and $D$ are fitting parameters. The binned covariances and the fitting curve are shown in
**Fig.15.**


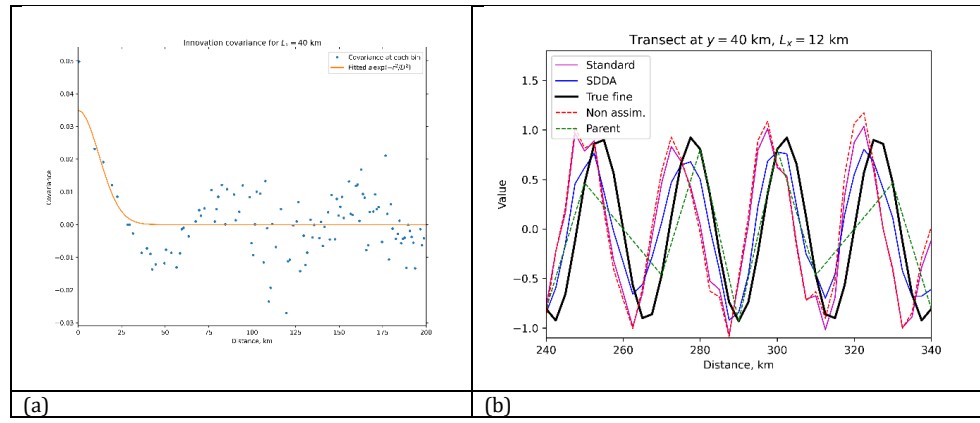

(a)                                    (b)


***Fig. 15.** (a) Estimates of background and observational error covariances for multiple eddy field with*
*$L_x = 40$ km and $L_y = 105$ km and the corresponding fitting curves. The error variances estimated at*
*r=0 are used as diagonal elements of the background matrix **B** (equal to 0.031) and the*
*'observational' matrix **R** (equal to 0.016 ), the length scale D=17km; (b) the zonal transect at $y =$*
*40 km showing the true field (black line), parent model (dashed green), child model noisy forecast*
*before DA (dashed red), analysis state after 'standard' DA (magenta), and analysis after SDDA (blue).*

Despite the parent model is assumed 'perfect', i.e. it has no errors relative to the true field, the H-L
method gives the error variance of approximately 50% of the noisy child model value. The **B** matrix is
then constructed using a variance (the diagonal) equal to 0.031 (see Fig 15 (a)) and the Gaussian



formula Eq (16), and the diagonal matrix **R** is build using a value for its variance (diagonal   elements)
of 0.016.
The **B** and **R** matrices were used to carry out the variational DA cycle to the simulated child model
forecast with the following parameters: multi eddy field with $L_x = 12$ km, $L_y = 105$ km, random
noise with STD=0.15, bias=0.3, spatial shift=4 km to the west. The analysis state is estimated using the
following equations of the 'standard' method (Kalnay, 2003, page 155)
$$x^a = x^b + \mathbf{W}[y - H(x^b)] = x^b + \mathbf{W}d$$
$$\mathbf{W} = (\mathbf{B}^{-1} + \mathbf{H}^T\mathbf{R}^{-1}\mathbf{H})^{-1}\mathbf{H}^T\mathbf{R}^{-1}$$
The transect in Fig.15(b) shows the analysis state for eddies with $L_x = 12$ km and $L_y = 105$ km
obtained by the 'standard' method together with the true solution, parent model output, noisy child
model forecast (before DA) and, for comparison, the analysis state obtained by the SDDA method.
The bias and RMSE relative to the true solution for the child model forecast and the analysis state
after 'standard' and SDDA data assimilation process is shown in **Table 1**
**Table 1.** Errors in the child model outputs before and after Data Assimilation

|       | Forecast (before DA) | 'standard' DA | SDDA  |
|-------|----------------------|---------------|-------|
| Bias  | 0.300                | -0.0066       | 0.000 |
| RMSE  | 0.608                | 0.5268        | 0.250 |


Fig. 16 shows the map of the analysis state produced by the SDDA and the standard methods.  The
errors are removed more efficiently in the areas of low values of the field variable $F$.

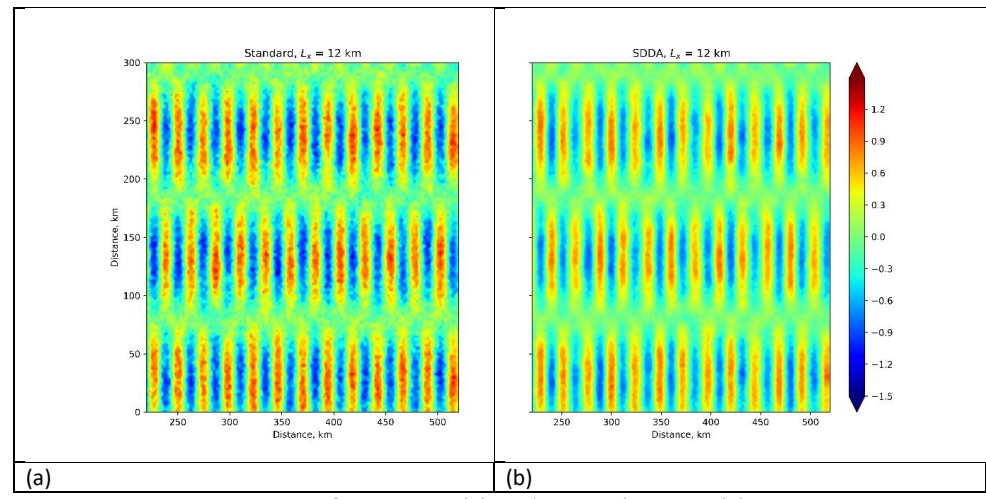

***Fig. 16.** The child model output after DA using (a) the 'standard' method, (b) the data assimilation*
*with Stochastic-Deterministic Downscaling (SDDA). The SDDA produces less noise.*

The analysis presented above shows that the SDDA method is not the only one which can be applied
to model-to-model data assimilation, the 'standard' method also gives reasonable results. However,
when compared with the 'standard' DA method, the SDDA gives better accuracy- stronger reduction



in the RMS errors and a complete removal of bias. The SDDA also is more computationally efficient.
We had no restrictions or limitations in computing the analysis for 1000×1000 km domain at 2.5 km
resolution when using the SDDA on our office PC. However, we were not able to carry out the
'standard' DA for a domain greater than 300×300km on the same PC due to computing resource
restrictions. In terms of speed of calculations, it took about 2 minutes to complete one full DA cycle
including calculation of the covariance matrix and weighting coefficients using the SDDA method in
the multiple eddy example. On the other hand, the standard method took 24 minutes, including the
creation of **B** and **R** matrices and calculation the analysis state.
Conclusion
This paper suggests a data assimilation approach where the data are assimilated into a high-resolution
model from a coarser good quality data assimilating model, not directly from observations. An
efficient and simple algorithm for model-to-model variational data assimilation method named Data
Assimilation with Stochastic-Deterministic Downscaling (SDDA) is developed. The theoretical
background behind the SDDA algorithm is discussed, and its application is illustrated in a number of
idealised synthetic situations which resemble real world practice in fine-resolution ocean modelling.
The results demonstrate that the model-to-model data assimilation is an efficient way of improving
the accuracy of fine resolution model. Such approach allows to avoid a repetition of a complex and
resource-hungry assimilation of actual observations which has already been done in the parent model.
It is likely that the same basic idea of model-to-model data assimilation would work also for other
methods currently used in observational data assimilation. In this paper, the SDDA was compared with
a commonly used Hollingsworth-Lönnberg method and shown to be more accurate and
computationally significantly less expensive.
Code and data availability
For code availability please contact the corresponding author.
Author contributions
GIS conceptualised and designed the study, developed the methodology,
contributed to software development, performed the analysis, and drafted the
paper. JMGO contributed to software development, analysis of the results, selection
of appropriate parameters and writing of the paper.
Competing interests
The authors declare that they have no conflict of interest.
Acknowledgements
The authors are thankful to the members of Plymouth Ocean Forecasting Centre for
their support.

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
