# Peer review of "Model-to-model data assimilation method for fine resolution ocean modelling"

_Ocean Science, 2021_

## Referee Comment (RC1)

**Review of "Model-to-model data assimilation method for fine resolution ocean modelling" by Georgy I. Shapiro and Jose Maria Gonzalez-Ondina.**

**General comments**

The manuscript presents a novel method for DA in a high-resolution ocean model (child) which is nested into a coarse-resolution data assimilated model (parent). The method consists of two steps: stochastic downscaling of the parent model to the grid of the child model, and an assimilation step using the downscaled field as observations and applying a simplified Kalman gain formula to obtain an analysis for each grid node independently. The method is demonstrated for 3 synthetic cases. The manuscript is well-written and nicely structured. The readability of the manuscript could however be improved by better visualization of the results.

My main concern with this manuscript is how well suited the three cases are to demonstrate the potential of the method for the proposed application, namely high-resolution ocean models. All three examples are two-dimensional fields of one state variable that can be described by smooth functions. I find the lack of small-scale variations that would mimic sub-mesoscale features in the true fields to be an unrealistic assumption in the presented experiments, as such variations will always be present in any realistic case. High-resolution models can be useful applications for dynamically active regions offshore. However, high-resolution models are more frequently applied for coastal regions. The examples in the manuscript do not address the suitability of the proposed method for regions where increased resolution is applied to better resolve complex topography and coastlines with mismatch in land mask between parent and child models, nor is this issue addressed in the discussion.

Another question that remains unanswered is whether the description of the ocean state is dynamically consistent after an assimilation procedure that is applied point-wise for each state variable. If this is not the case, it will most likely result in numerical instabilities when initializing the forecast from such an analysis and thus additional post-processing will be required for the method to be applicable.

While the motivation of the preprint is highly relevant and the suggested method has some appealing qualities, I am not convinced that the results presented here are sufficient to support the claim that the SDDA is an appropriate method for improving the accuracy of a fine resolution ocean model. Therefore, I recommend reconsidering publishing the manuscript after major revision.

**Specific comments**

**Introduction:**
*Line 63-64:* The synthetic cases are not introduced or otherwise mentioned in the main body of the manuscript before this point. I would suggest either rephrasing or including a sentence or two in the above paragraph.

**Data and Methods:**

*Line 80:* In the general equation for the cost function the observation operator $H$ may include a conversion from model variable to the observed quantity in addition to interpolation, although this is not the case for the presented method.

**Results:**

$\mathbf{V}_B$ and $\mathbf{V}_R$ determine how much weight is given to the fluctuations of the background vs the fluctuations of the downscaled product. I cannot see that there's any mention of how the values of these key parameters are set, nor a discussion on how these choices affect the results. The aim should be to weight the two solutions in a way where the child model is prevented from drifting from the assimilated parent model, while at the same time retaining high-resolution dynamical features that arise from e.g. improved topography.

The proposed SDDA method consists of two steps, namely a stochastic downscaling of the parent model to the grid of the child model, and an assimilation step where the downscaled parent model values are treated as observations and combined with the first guess of the child model. I think an analysis of how these two steps contribute to the total improvement of the SDDA could provide valuable insights. Given the smooth nature of the chosen examples, I suspect the DA step might increase RMSE and bias compared to the intermediate solution given by the SD step.

*Fig 7:* It is nearly impossible to see any difference between figures 7c and 7d - perhaps zooming in on a smaller part of the grid or using a different colormap for the difference plot could be beneficial?

**Discussion:**
In the discussion, the SDDA method is compared with a "standard" DA method, and is demonstrated to be superior. Again, the choices for $\mathbf{B}$ and $\mathbf{R}$ will strongly affect the results and it would thus be relevant to report whether or not they differ significantly between the two methods, as well as how changing their values affect the results. Evaluating the cost function values in addition to RMSE and bias could perhaps also help to shed light on the differences.

As spectral nudging (see e.g. Katavouta and Thompson, 2016) addresses the very same issue as the proposed method, namely ensuring that a high-resolution model does not drift away from the large-scales that are well constrained by observations assimilated in coarser ocean models, I think a discussion on how the SDDA method compares with spectral nudging both in terms of quality of the results and computational efficiency would be of interest to the target audience for this manuscript.

**Figures:**
The authors might want to consider adopting scientific colormaps and avoid using red and green colored lines in the same plots. See e.g. Crameri et al. (2020).

**Technical comments**

**Data and methods:**
*Equation 1:* the observation vector $\mathbf{y}$ should be in bold font.
*Line 84:* A period is missing after the reference.
*Line 96:* a "b"-superscript seems to be missing from the left-hand side of the equation.

*Line 200 - 201:* Same resolution is stated for both parent and child.
*Line 214:* "A = 1"
*Line 219:* "eddies *can* exist nearly anywhere"

**Results:**
*Line 241:* "an analysis" or perhaps initial conditions?
Although the section indeed presents results for cases A-D, the statement at line 242 of "four examples" reads as a typo in the current context.

**Discussion:**
*Line 424:* observation***s***.
*Line 468:* mode***l***-to-model

**References**

Crameri, F., Shephard, G.E. & Heron, P.J. The misuse of colour in science communication. Nat Commun 11, 5444 (2020). https://doi.org/10.1038/s41467-020-19160-7

Katavouta, A., and Thompson, K. R. (2016). Downscaling ocean conditions with application to the gulf of maine, scotian shelf and adjacent deep ocean. Ocean Model. 104, 54–72. doi: 10.1016/j.ocemod.2016.05.007

---

## Referee Comment (RC2)

**Review of "Model-to-model data assimilation method for fine resolution ocean modelling" by Shapiro and Gonzalez-Ondina.**

The paper describes a method to combine information from a high-resolution ocean model forecast with a lower resolution assimilating model forecast, building on a method called Stochastic-Deterministic Downscaling presented in a previous paper by Shapiro et al. (2021). This fits well with the journal topics, and the work presents application of a novel method. The method is demonstrated using some very idealised cases where the true field is known and a lower resolution version of the true field is combined with a high resolution version with random noise added. In these cases the SDDA method is shown reduce the error in the analysis. The SDDA method is also compared to a more conventional data assimilation approach.

There are a number of shortcomings in the paper including the simplified nature of the experimental set-up (no model is actually used so we can't tell how well the analysis would initialise a dynamical model and only one 2D variable is analysed); also the comparison to the "standard" data assimilation seems rather flawed since the set-up of the standard DA is very basic and is not described very clearly. These and other comments are listed below which should be addressed before publication so I recommend major revisions.

Comments on the motivation for the work and introduction

- The authors mention the availability of ocean models such as ROMS and NEMO, but do not mention the availability of data assimilation software such as DART, PDAF and others.
- I'm not sure from the description in the introduction if the issue being addressed by the new method is the need for expertise in DA, the lack of access to DA software, the lack of computational resources, or just that methods for improving the way high resolution models are initialised can be improved. This motivation could be made clearer.
- There seems to be a lack of a literature review on downscaling methods, for example the work of Katayouta and Thompson (2016) and von Storch et al. (2000). Generally the references to certain topics seem quite out of date.
- Line 67. The authors seem to be saying that variational methods and OI are the same which is not true.
- The overall structure of the paper seems appropriate but the section names, their titles and references to them in the text seem a bit confused sometimes.

Comments on the method

- I think it would be useful to clarify some aspects of the method in contrast to issues faced in standard DA. For instance, using the coarse model data as "observations" means that the spatial correlations in the coarse model errors need to be dealt with. One therefore needs to know the accuracy of the parent model which will depend on many factors including where observations were recently assimilated as well as the forecast error growth in the parent model. It is much more complicated to know the characteristics of the errors in the parent model than in the observations themselves.
- It would be good to address the question of whether the method aims to retain the spectral characteristics of the child model (e.g. as the spectral nudging method of Katayouta and Thompson)?

- Line 81: B is the background or forecast error covariance. The authors say "model" error covariance which is something different.
- Line 84-86: These statements aren't specific enough. The size of matrix B is not affected by the number of observations. It is normally very large though, and methods have been developed to represent approximations of it efficiently. The size of matrix R is dependent on the number of observations, and most assimilation systems assume R to be diagonal (no observation error correlations) so that its inverse can be quickly/easily calculated.  That approach is obviously not appropriate in the case where the "observations" coming from the parent model will have significant correlations.
- The statement on line 120-123 seems crucial to me. Clearly the high-resolution system does not have the same correlations as the coarse model (e.g. if one includes sub-mesoscale processes and one doesn't then they would have very different correlations). If the downscaling operator S allows the "observations" S(y) to contain the same spatial correlations as the high-resolution model then this could be justified, but this needs some further comments and evidence to justify it. There is some discussion on this starting line 142 and the reader is referred to the separate paper on the SDD method.
- How would the method deal with islands in the child model which are not in the parent model, or different coastlines/bathymetries?
- I found it difficult to understand what the mean is referring to. Line 150-153 seems to be saying that ideally an ensemble would be used to generate an ensemble mean, but that instead a local spatial mean is used. Perhaps that could be made clearer.
- Line 157: the error variance of the mean could be obtained if there was an ensemble, e.g. using the ensemble spread.
- Line 160: setting the mean from the child model with the mean from the parent model seems problematic. Won't this remove the high-resolution spatial structure in the mean of the child model? In the case of a front, the high-resolution system would presumably have higher gradients in the mean compared with the parent model, and this benefit from having the child model will be removed.
- It would be useful to have a more stand-alone description of the SDD method which is the crux of the method used in this paper. At the moment the reader has to read a different paper to really understand what is being done.
- Line 188: A more recent reference is Janjic et al 2018 where they recommend using the term "representation error".

Comments on the experimental set-up

- Line 191: Only random noise is added to the parent model. In reality the parent model will contain errors which have spatial and temporal correlations and most likely biases.
- Only random noise is added to the child model which seems to be to represent features not resolved in the parent model. This seems rather unrealistic since the extra features in the child model will have structure, e.g. associated with sub-mesoscale processes.
- Line 201: this is a typo as it says the child model is at the same resolution as the parent model.
- Line 202: A correlation function is mentioned. How is the correlation function defined?  How is it modelled? Why was it set to zero beyond some distance?
- Line 203: why was a region of 68 sq km chosen for the spatial averaging? Is there some justification for it, or analysis of the sensitivity of results to it?
- Line 214: Typo: is it supposed to be A=1?
- Eq (14): how are x and y defined in this equation?
- Line 232: The eddy size in the y direction of 105 km seems large compared to the range of sizes in the x direction. Is this justified?
- There doesn't seem to be any model involved in the experiments. So how do the experiments relate to the title of the paper and the introduction which give the impression that this work is done in the context of assimilating data into a high-resolution model.

- What is the noise in the child model forecast meant to represent? In practice the child model will have errors of course, but it is unlikely they will be uncorrelated white noise.

Comments on the results

- Line 250: this is the first mention of adding bias I think. This should be mentioned in the experimental set-up section.
- Line 256-258: This statement seems strange. The field is constant away from the front so it wouldn't be hard to represent it.
- Line 261: this stand-alone sentence ought to be attached to the subsequent paragraph.
- Figs 2 & 3: There's a lot of noise in the analysis. I would have thought if the error variances were specified correctly that this noise would be largely removed. How do you estimate the error variance of the S(y)?
- Fig 4(b): the caption doesn't mention the red line.
- Fig 5: You don't show the true field on the parent grid in this example which would be nice to see.
- Fig 6: why does the RMSE start rising again for eddy sizes > about 25 km?
- Line 366: "…random noise in the non assimilated model". Not sure what you mean there.
- Fig 11: The green line seems to indicate that the method can't deal very well with random noise.
- Generally the method seems to be dealing well with reducing the bias which is calculated over the whole domain so that positive/negative differences will average out.
- Line 401: "the improvement is five to ten fold". Where do these numbers come from? They don't seem consistent with the ratio line in Fig. 12 (b).

Comments on the discussion and conclusions

- Line 437: you should include references for these methods for error covariance estimation and also reference the review by Bannister (2008).
- Line 438-439: I don't understand this sentence.
- Fig 14: The amplitude of the peak seems to be reduced in the SDDA method which should be mentioned.
- Line 467: It seems like this section comparing the SDDA method to the standard assimilation warrants a section of its own rather than being in the discussion section. The standard data assimilation chosen is only one of many available and the authors seem to have simplified its application significantly. Also, the SDDA method relies on the standard method producing a very good quality parent model solution.
- Line 478: In most applications I know of, the model and observation error variances are not assumed to be homogeneous as is stated here. They would be estimated as spatially varying fields. If this was done, how would your conclusions be affected?
- Line 497: Where were the observations located? Or did you assume observations to be available everywhere?
- Fig. 15(a). The quality of the figure is very poor and I found it hard to see the numbers or read any of the text. I couldn't tell where the observation error covariance was plotted as stated in the caption.
- Eq on line 525-526: K is usually used as the notation for the Kalman gain matrix.
- Fig 16: Could you show also the error (compared to the truth) in the standard method and the SDDA method?

- Line 542: It is stated that SDDA is more computationally efficient. Wouldn't it be fairer to compare the standard method with the total cost of assimilating the data into the parent model and then doing the SDDA?

References:

- Katavouta and Thompson, 2016. Downscaling ocean conditions with application to the Gulf of Maine, Scotian Shelf and adjacent deep ocean. Ocean Model., 104 (2016), pp. 54-72, 10.1016/j.ocemod.2016.05.007
- von Storch, H., Langenberg, H., & Feser, F. (2000). A Spectral Nudging Technique for Dynamical Downscaling Purposes, Monthly Weather Review, 128(10), 3664-3673.
- Janjić, T, Bormann, N, Bocquet, M, Carton, JA, Cohn, SE, Dance, SL, Losa, SN, Nichols, NK, Potthast, R, Waller, JA, Weston, P. On the representation error in data assimilation, Q J R Meteorol Soc. 2018; 144: 1257– 1278. https://doi.org/10.1002/qj.3130

---

## Author Comment (AC1)

Authors' responses to Reviewer 1 comments on the MS 'Model-to-model data assimilation method for fine resolution ocean modelling' by Shapiro and Ondina

**Comment.** *The manuscript presents a novel method for DA in a high-resolution ocean model*

**Response.** Thank you

**Comment.** *My main concern with this manuscript is how well suited the three cases are to demonstrate the potential of the method for the proposed application, namely high-resolution ocean models. All three examples are two-dimensional fields of one state variable that can be described by smooth functions.*

**Response.** The two-dimensional fields used in the MS represent data from a single computational (geopotential or sigma) level taken from a full 3D mesh. In order to carry out DA in full 3D the process has to be repeated for all levels. Such approach is widely used in practical applications of DA in ocean and atmospheric modelling. For example, the papers referenced in our original MS (Adhikary el al. 2008) and (Bell et al. 2000) present schemes where forecast error correlations only include horizontal dependencies and DA is performed level-by-level in the same way that is used in our method. Clarification has been given in the text.

**Comment.** *I find the lack of small-scale variations that would mimic sub-mesoscale features in the true fields to be an unrealistic assumption in the presented experiments, as such variations will always be present in any realistic case.*

**Response**. Whether an ocean feature is mesoscale or sub-mesoscale depends on the Rossby radius (commonly the first baroclinic radius is used)- see (Robinson, 1983). If the Rossby radius is about R=50 km (as in the mid Atlantic) or even 10 km (as in some coastal areas, e.g. the Persian Gulf) then the eddies of 6 km in radius, treated in the Example b) of the MS are definitely sub-mesoscale. The eddies of 46 km radius again considered in Example b) could be classed as mesoscale at R=50 km. The minimum size of sub-mesoscale features resolved by a model is determined by the resolution of the child (fine) model not by the SDDA methodology. The SDDA method and the examples in the MS are not specific to a certain value of Rossby radius, and therefore can be applied to both meso and sub-meso features. In order to clarify this issue, we have added the relevant explanation.

**Comment.** *The examples in the manuscript do not address the suitability of the proposed method for regions where increased resolution is applied to better resolve complex topography and coastlines with mismatch in land mask between parent and child models, nor is this issue addressed in the discussion.*

**Response.** A potential mismatch between fine and coarse grids due to finer features of the coastline and bathymetry can only relate to the first step of the SDDA method, namely the downscaling. This issue is not specific to SDDA and may appear when gridded observational data of different resolution (e.g. satellite imagery) is used in common data assimilation procedures. The downscaling of the coarse model data onto the fine model grid is carried

out using the SDD algorithm. As shown in (Shapiro et al, 2021) in the example of the coastal areas of the Red Sea, this mismatch is natively resolved during the SDD downscaling. Better resolution of the coastline/bathymetry could result in higher vorticity/enstrophy values in the downscaled field compared to the parent coarse model as shown in Fig 11-15 of the above paper. After downscaling the data from the parent and child model are available exactly on the same (fine) grid, and the grid mismatch issue does not appear. Additional reference and clarification are given in the revised MS.

**Comment.** *Another question that remains unanswered is whether the description of the ocean state is dynamically consistent after an assimilation procedure that is applied point-wise for each state variable. If this is not the case, it will most likely result in numerical instabilities when initializing the forecast from such an analysis and thus additional post-processing will be required for the method to be applicable.*

**Response.** This issue is common to a variety of data assimilation methods, it is not specific to assimilating data from another model instead of observations. In a wider context, hydrostatic instability could appear when external data are incorporated into the model. An example is the initialisation of numerical model from climatological fields of temperature and salinity when separate interpolation of the state variables to the model grid may result in inversion in density. The issue is well known and is dealt with in a number of ways. For example, in the NEMO model, density inversions are treated with highly enhanced vertical diffusion, so that the inversions are removed in a few time steps. Clarification is added to the text.

**Comment.** *Line 63-64: The synthetic cases are not introduced or otherwise mentioned in the main body of the manuscript before this point. I would suggest either rephrasing or including a sentence or two in the above paragraph.*

**Response.** The text amended as advised.

**Comment.** $V_B$ *and* $V_R$ *determine how much weight is given to the fluctuations of the background vs the fluctuations of the downscaled product. I cannot see that there's any mention of how the values of these key parameters are set, nor a discussion on how these choices affect the results. The aim should be to weight the two solutions in a way where the child model is prevented from drifting from the assimilated parent model, while at the same time retaining high-resolution dynamical features that arise from e.g. improved topography.*

**Response.** Diagonal matrices $V_B$ and $V_R$ are related to the error covariance matrices $B$ and $R$ of the child and parent models respectively as specified in lines 114-115 of the original MS. Therefore, the values of these key parameters are not set by a modeller, i.e., they are not tuning coefficients but calculated using an algorithm similar to other variational DA methods. This point is briefly discussed in the original MS (lines 436-438). As both data sets are from models (the parent model outputs are used instead of real observations) then any suitable method to calculate the matrix $B$, can be applied to calculate matrix $R$, and hence $V_B$ and $V_R$, for example the NMC (Parrish, D. F. and Derber, J. C., 1992: The National Meteorological Center's spectral statistical interpolation analysis system. Mon. Weather Rev., 120, 1747-1763 ) or 'Canadian' (Polavarapu et al, 2005, Data assimilation with the Canadian middle atmosphere model, Atmosphere-Ocean, 43:1, 77-100, DOI: 10.3137/ao.430105 ) methods. In the examples presented in the MS, the diagonal elements $V_{Bii}$ and $V_{Rii}$ ( see Eq 10) required for the second step of SDDA are calculated, for consistency, in the same way as for the first step (downscaling) , namely of by spatial averaging and calculating

dispersion of fluctuations over a small trial area around the node at the same time point, see lines 152-153 of the original MS which refers the reader to (Shapiro et al, 2021) for details. The trial area was a square of 68 x 68 km centred at each node (see lines 203-204 of the original MS). We agree that it would be easier for a reader if such details are presented in greater detail in the actual MS. The text of the revised MS is extended to incorporate this and a new figure (see below) showing an example map of VBii and $V_{Rii}$ is added, along with the clarifying text .

[Figure]

**Comment.** *The proposed SDDA method consists of two steps, namely a stochastic downscaling of the parent model to the grid of the child model, and an assimilation step where the downscaled parent model values are treated as observations and combined with the first guess of the child model. I think an analysis of how these two steps contribute to the total improvement of the SDDA could provide valuable insights. Given the smooth nature of the chosen examples, I suspect the DA step might increase RMSE and bias compared to the intermediate solution given by the SD step.*

**Response.** An additional analysis is provided in the discussion section of the revised MS as advised.

**Comment.** *Again, the choices for **B** and **R** will strongly affect the results and it would thus be relevant to report whether or not they differ significantly between the two methods, as well as how changing their values affect the results.*

**Response.** The **B** and **R** matrices are not prescribed (chosen) but calculated according to the SDDA and H-L methods accordingly, so there is no option to change them arbitrarily in order to see how their values affect the results. The procedures for calculation of **B** and **R** matrices in the SDDA and H-L methods are different and therefore they produce different matrices. For SDDA, both matrices represent the covariances of the background error of the downscaled parent and child models respectively and are calculated in a similar and consistent way as both data sets are from models. Therefore, whilst both matrices have different error variances (diagonal elements), they have the same spatial correlations. Conversely, in the 'standard' method, the matrices are calculated using the H-L method that assumes that the observation errors have no spatial correlation. This assumption can be valid for some types of observations, but it is not valid for model-to-model DA.

The revised MS is amended to include the discussion of differences in **B** and **R** matrices for the two methods as advised.

**Comment.** *Evaluating the cost function values in addition to RMSE and bias could perhaps also help to shed light on the differences.*

**Response.** The cost functions for the 'standard' (see Eq (1)) and the SDDA (see Eq (2)) methods are different, so it is not clear what information can be gained from the direct comparison. Both methods minimize their own cost functions. Clarification is given in the revised MS.

**Comment.** *As spectral nudging (see e.g. Katavouta and Thompson, 2016) addresses the very same issue as the proposed method, namely ensuring that a high-resolution model does not drift away from the large-scales that are well constrained by observations assimilated in coarser ocean models, I think a discussion on how the SDDA method compares with spectral nudging both in terms of quality of the results and computational efficiency would be of interest to the target audience for this manuscript.*

**Response.** The study by  Katavouta and Thompson (2016) used a spectral nudging method in order to restrict the drift of the fine scale model from the global model. They described a method that 'nudges' the large scale spectral components of a regional model to those of a global model. The main difference with the SDDA is that the nudging technique uses weighting coefficients that are tuning parameters and are prescribed in advance, while the SDDA variational method uses weights computed from the variance of the errors by minimising the cost function and therefore they cannot be changed at will. In contrast to the spectral nudging, the SDDA method corrects both large and small scale components of the child model as seen in amplitude spectra shown in Fig.14 of the original MS. The removal of bias in the Fourier space can be seen in Fig. 14 of the original MS. Some similarity can be found in the fact that the bias  of the child model is replaced with the bias from the parent model which could be interpreted as  aggressively nudging of a single long-wave component of the field. The relevant reference and discussion are added in the revised MS as advised.

**Comment.** *The authors might want to consider adopting scientific colormaps and avoid using red and green colored lines in the same plots. See e.g. Crameri et al. (2020).*

**Response.** The green colour is replaced with a different shade in line-art plots in the revised MS as advised. Different markers have been also added to improve readability of plots with many different lines. The jet colour map which contains both red and green colour is used for compatibility with the EU Copernicus Marine Service products, and other recent papers published in Ocean Science, where maps contain both green and red,  e.g. doi.org/10.5194/os-17-1385-2021 ; doi.org/10.5194/os-17-833-2021; doi.org/10.5194/os-17-615-2021 etc.

**Comment.** *Equation 1: the observation vector **y** should be in bold font*

**Response.** Corrected as advised.

**Comment.** *Line 84: A period is missing after the reference*

**Response.** Corrected as advised.

**Comment.** *Line 96: a "b"-superscript seems to be missing from the left-hand side of the equation.*

**Response.** Corrected as advised.

**Comment.** *Line 241: "an analysis" or perhaps initial conditions?*
**Response.** The MS considers only one assimilation cycle, therefore the 'analysis' of the previous cycle acts as initial condition for the next cycle. Clarification is given.

**Comment.** *Although the section indeed presents results for cases A-D, the statement at line 242 of "four examples" reads as a typo in the current context.*

**Response.** The sentence is corrected to include all 4 examples as advised

**Comments.** *Line 424: observations.*
*Line 468: model-to-model*

**Responses.** Both typos corrected as advised.

---

## Author Comment (AC2)

Author's responses to reviewer 2 comments on the MS 'Model-to-model data assimilation method for fine resolution ocean modelling' by Shapiro and Ondina

**Comment.** *This fits well with the journal topics, and the work presents application of a novel method.*

**Response** Thank you

**Comment.** *There are a number of shortcomings in the paper including the simplified nature of the experimental set-up*

**Response.** The purpose of this paper is to introduce a novel method and illustrate its quality in a synthetic case where the true solution is known. One of the problems in data assimilation is that the true solution is not known and therefore the error covariance matrices required to minimise the cost function are estimated based on various assumptions, see e.g. the review paper by Bannister (2008). Comparison with observations has its own drawbacks as observations are not perfect. Therefore, testing the new method in situations when the true solution is known should be considered as an advantage not shortcomings. This approach is not new, the testing of a DA method on a synthetic case was described e.g. in the (Carrassi et al, 2018) , and we followed this approach. Clarification is given in the text.

**Comment.** *Also the comparison to the "standard" data assimilation seems rather flawed since the set-up of the standard DA is very basic and is not described very clearly.*

**Response.** The procedure for the 'standard' data assimilation closely follows the E.Kalnay's data assimilation textbook to which a reference is given. The interested reader is directed to this book for details. In using a simplified synthetic case we followed the practice described in (Carrassi et al, 2018), see our response to the previous comment for more information.

**Comment.** *The authors mention the availability of ocean models such as ROMS and NEMO, but do not mention the availability of data assimilation software such as DART, PDAF and others.*

**Response.** Additional references are added as advised.

**Comment.** *I'm not sure from the description in the introduction if the issue being addressed by the new method is the need for expertise in DA, the lack of access to DA software, the lack of computational resources, or just that methods for improving the way high resolution models are initialised can be improved. This motivation could be made clearer.*

**Response.** The existence of DA software does not mean that the DA is a simple process. The new method is aimed at small groups ( sometimes in developing countries) who would like to run high-resolution local ocean models, have limited computational resources but have access to freely available outputs from global/ basin scale models such as provided by EU Copernicus Marine Environment Monitoring Service (CMEMS) . The EU modelling community encourages the downscaling of CMEMS products by end users. As far as we know such groups just download or downscale (by nesting) the products from CMEMS but it is too difficult for them to use the complex

DA methods in the same way as in the world leading ocean forecasting centres. The SDDA method is simple and computationally efficient enough and would help these groups to use DA. For the period of 1.5 month up to 7 October, there were 353 views of the preprint of the paper registered by Ocean Science website which indicate that there is sufficient audience for the new method. The text is amended to clarify the motivation.

**Comment.** *There seems to be a lack of a literature review on downscaling methods, for example the work of Katayouta and Thompson (2016) and von Storch et al. (2000). Generally the references to certain topics seem quite out of date.*

**Response.** As in any research (not review) paper the number of references is limited. Following the reviewer's advice we have added a reference to Katavouta and Thompson (2016) as well as clarification of the differences between spectral nudging used in the above paper and the SDDA method. We prefer to give references to the original papers, and some classic results were obtained some time ago, for example the H-L method (1986) or Lorenc (1986) etc. This is not something new. The excellent paper by Bannister (2008) goes as far back in its reference list as the year of 1954.

**Comment.** Line 67. The authors seem to be saying that variational methods and OI are the same which is not true.

**Response.** The reviewer is not correct here. Lorenc (1986) showed that the OI solution is equivalent to a specific variational assimilation problem: Find the optimal analysis $x_a$ field that minimizes a (scalar) cost function- see the original study by Lorenc which states: '"Not surprisingly therefore, Wahba and Wendelberger (1980) have shown that OI can also be expressed in terms of the solution of a variational problem." More recent confirmation of this fact can be found in (http://www.atmos.berkeley.edu/~inez/MSRI-NCAR_CarbonDA/lectures/Kalnay3_Ch53D_Var.pdf; http://twister.caps.ou.edu/OBAN2019/3DVAR.pdf etc). The OI is also known as the Best Linear Unbiased Estimator (BLUE), see e.g. https://docs.salome-platform.org/latest/gui/ADAO/en/ref_algorithm_Blue.html

**Comment.** *The overall structure of the paper seems appropriate but the section names, their titles and references to them in the text seem a bit confused sometimes*

**Response.** From this comment it is not clear what exactly is confusing in the titles of section and what is the difference between the 'names' and the 'titles'.

**Comment.** *I think it would be useful to clarify some aspects of the method in contrast to issues faced in standard DA. For instance, using the coarse model data as "observations" means that the spatial correlations in the coarse model errors need to be dealt with. One therefore needs to know the accuracy of the parent model which will depend on many factors including where observations were recently assimilated as well as the forecast error growth in the parent model. It is much more complicated to know the characteristics of the errors in the parent model than in the observations themselves.*

**Response.** In contrast to the standard DA (e.g H-L method), no assumption is made that the 'observations' are not spatially correlated. Actually, the data in the parent model are well correlated

as they are quite densely spaced. We agree that the accuracy of the parent model is an important factor. It is obvious that assimilating bad data would not improve the high-resolution model. In order to clarify this point, the MS considers a few examples (a to c) where the parent model is error-free (except errors caused by missing data due to insufficient resolution) and an example (d) where the parent model is noisy. As we know the true solution, we can estimate the errors i.e deviation from the known true solution of any data set: parent model, downscaled parent model, child model before and after SDDA. The knowledge of the true solution allows calculating the improvement of the child model output after the full SDDA cycle. This is done both in cases of error-free and noisy parent model. The SDDA method is aimed at a large community of users who utilise the outputs from global or basin scale models such freely available from CMEMS. Therefore, they trust that such products have the possible accuracy. The quality of CMEMS products is frequently monitored in Copernicus Marine Environment Monitoring Service (CMEMS) Ocean State Reports. A discussion is added as well as clarification on how much improvement comes from the first (SDD) step and how much from the second (DA) step.

**Comment.** *It would be good to address the question of whether the method aims to retain the spectral characteristics of the child model (e.g. as the spectral nudging method of Katayouta and Thompson)?*

**Response.** The answer to this question is given in Fig 14 of the original MS which shows the amplitude spectra of the child model before and after SDDA. The SDDA method does not retain the spectral characteristics of the non-assimilated child model in full. The SDDA removes the bias (which can be seen as a very long-wave component), this is somewhat similar to spectral nudging but more aggressive. Our model attempts to retain the "bias" of the coarse model, which in practice is a local spatial average with a length scale of the order of the correlation length scale. In contrast to spectral nudging by Katayouta and Thompson, the spectral components of noise are removed both in high-frequency and low frequency range, therefore the spectral component of the true signal is represented better after the SDDA. The left panel of Fig14 is reproduced here for convenience. The text is amended to clarify this issue.

[Figure]

**Comment.** *Line 81: B is the background or forecast error covariance. The authors say "model" error covariance which is something different.*

**Response .** Thank you, the text is corrected as advised.

**Comment.** *Line 84-86: These statements aren't specific enough. The size of matrix B is not affected by the number of observations. It is normally very large though, and methods have been developed to represent approximations of it efficiently.*

**Response .** We agree. However, no matter how good these methods of approximation are, it is still true that as B and R grow in size, the problem becomes larger and requires more time and computer resources to be solved. These time and resources are a significant barrier for many small groups. The text is corrected accordingly.

**Comment.** *The size of matrix R is dependent on the number of observations, and most assimilation systems assume R to be diagonal (no observation error correlations) so that its inverse can be quickly/easily calculated. That approach is obviously not appropriate in the case where the "observations" coming from the parent model will have significant correlations*

**Response.** This is correct. Therefore, the SDDA method does not assume that the R matrix is diagonal.  Clarification is given in the text.

**Comment.** *The statement on line 120-123 seems crucial to me. Clearly the high-resolution system does not have the same correlations as the coarse model (e.g. if one includes sub-mesoscale processes and one doesn't then they would have very different correlations). If the downscaling operator S allows the "observations" S(y) to contain the same spatial correlations as the high-resolution model then this could be justified, but this needs some further comments and evidence to justify it. There is some discussion on this starting line 142 and the reader is referred to the separate paper on the SDD method*

**Response.** This is true. The error covariance matrix B (for the child model) may be different from the error covariance matrix R (for the parent model) at very small distances not resolved by the parent model. It has to be noted that in any DA method, model error covariance matrices are not exact and are estimated based on using various approximations. In his excellent review paper, R. Bannister (2008) states 'Owing partly to its very large rank, the **B**-matrix is impossible to use in an explicit fashion in an operational setting and so methods have been sought to model its important properties in a practical way… It is also common to assume that the **B**-matrix is static'. Whatever the approximation for the **B** (and **R**) matrix is used, its quality can be seen in how well the DA reduces the model errors. The SDDA method is shown to significantly reduce both bias and the RMSA, see Figs 2 to 14 of the original MS. It is also shown that the assumption of a similar spatial correlation structure between **B** and **R** matrices used in SDDA produces better results that the assumption of a non-correlated **R** matrix as used in the widely used H-L method, see Table 1 of the original MS. Clarification and a reference to Bannister (2008) is now added in the revised MS.

**Comment.** *How would the method deal with islands in the child model which are not in the parent model, or different coastlines/bathymetries?*

**Response.** A potential mismatch between fine and coarse grids due to finer features of the coastline and bathymetry can only relate to the first step of the SDDA method, namely the downscaling. The downscaling of the coarse model data onto the fine model grid is carried out using the SDD algorithm. As shown in (Shapiro et al, 2021) in the example of the coastal areas of the Red Sea, this mismatch is natively resolved during the SDD downscaling. Better resolution of the coastline/bathymetry could result in higher vorticity/enstrophy values in the downscaled field compared to the parent coarse model as shown in Fig 11-15 of the above paper. After downscaling the data from the parent and child model are available exactly on the same (fine) grid, therefore there is no grid mismatch at the DA step. Additional reference and clarification are given in the revised MS

**Comment.** *I found it difficult to understand what the mean is referring to. Line 150-153 seems to be saying that ideally an ensemble would be used to generate an ensemble mean, but that instead a local spatial mean is used. Perhaps that could be made clearer.*

**Response .** A local spatial mean as well as standard deviation of fluctuations are  calculated in the small trial area around each node. Clarification is given in the text.

**Comment.** *Line 157: the error variance of the mean could be obtained if there was an ensemble, e.g. using the ensemble spread.*

**Response.** We do not calculate the variance of the mean, only the variance of fluctuations around the mean. Clarification is given in the text.

**Comment.** *Line 160: setting the mean from the child model with the mean from the parent model seems problematic. Won't this remove the high-resolution spatial structure in the mean of the child model? In the case of a front, the high-resolution system would presumably have higher gradients in the mean compared with the parent model, and this benefit from having the child model will be removed.*

**Response.** The replacement of the mean from the child model with the mean from the parent model is equivalent to an aggressive spectral nudging when only the largest wavelength (corresponding to the mean within the trial area) is nudged. It does not remove or modify any higher frequency components. The higher frequency components are improved by the DA of fluctuations as represented by Eq (10). The removal of bias and improvement of other frequencies is seen on the amplitude spectra shown in Fig. 14 of the original MS.

**Comment.** *It would be useful to have a more stand-alone description of the SDD method which is the crux of the method used in this paper. At the moment the reader has to read a different paper to really understand what is being done.*

**Response.** The SDD method is described in detail in the paper which is easily accessible for free to a reader from the journal website, so there is no need to repeat that paper in the MS.

**Comment.** *Line 188: A more recent reference is Janjic et al 2018 where they recommend using the term "representation error".*

**Response.** We prefer to reference the original papers, or the review papers / textbooks where necessary instead of secondary more recent papers.

**Comment.** Line 191: *Only random noise is added to the parent model. In reality the parent model will contain errors which have spatial and temporal correlations and most likely biases.*

**Response.** The parent model is assumed to be of good quality and have small errors of whatever nature due to its own data assimilation procedure. These errors are spatially correlated as described in sub-section 'The algorithm' in particular see the part between Eq(4) and Eq(5) defining the correlation matrix $C_R$. The random noise is added to the parent model just as an illustration of the performance of the algorithm to the combination of correlated and non-correlated noise. If the parent model is of bad quality, it would not be prudent to use it for data assimilation, this is similar to rejection of bad observational data.

**Comment.** *Only random noise is added to the child model which seems to be to represent features not resolved in the parent model. This seems rather unrealistic since the extra features in the child model will have structure, e.g. associated with sub-mesoscale processes.*

**Response.** The noise in the child model represents its own errors (as every model has) not the real features not resolved by the parent model. The errors in the child model include both uncorrelated (random Gaussian noise) and spatially correlated (spatial shift) components in various combinations, see Figs 2,3 11 of the original MS. After SDDA the child model reveals the small scale features not resolved in the parent model, see for example Fig.10(a).

**Comment.** *Line 201: this is a typo as it says the child model is at the same resolution as the parent model*

**Response.** Thank you, the typo is corrected.

**Comment.** *Line 202: A correlation function is mentioned. How is the correlation function defined? How is it modelled? Why was it set to zero beyond some distance?*

**Response.** The correlation function is modelled by a Gaussian function in common with other DA methods. It is set to zero at distances greater than a double correlation radius as the value of Gaussian function beyond that distance is less than 2%. The spatial cut-off of the B matrix is common for DA methods (see e.g. Carrassi, 2018) . The details of computing the correlation matrix are given in (Shapiro et al, 2021).

**Comment.** *Line 203: why was a region of 68 sq km chosen for the spatial averaging? Is there some justification for it, or analysis of the sensitivity of results to it?*

**Response .** The trial area is set to 4Lx4L where L is the correlation radius. The Gaussian function has a value of less than 2% beyond that and its contribution can be safely neglected. Clarification is added to the text.

**Comment.** *Line 214: Typo: is it supposed to be A=1?*

**Response.** Thank you, the typo is now corrected.

**Comment.** Eq (14): how are x and y defined in this equation?

**Response**. The coordinates x and y are referenced to the centre of the domain in the zonal and meridional directions respectively. Clarification is added to the text.

**Comment.** Line 232: *The eddy size in the y direction of 105 km seems large compared to the range of sizes in the x direction. Is this justified?*

**Response.** The meridional and zonal sizes of features in example c) are intentionally different to reveal how SDDA performs in an anisotropic setting.

**Comment.** *There doesn't seem to be any model involved in the experiments. So how do the experiments relate to the title of the paper and the introduction which give the impression that this work is done in the context of assimilating data into a high-resolution model.*

**Response.** The SDDA algorithm is developed for use with a high-resolution deterministic (dynamic) child model run locally and combined with the outputs from a lower-resolution parent model which can be obtained either onsite or downloaded from respectful ocean forecasting services such as CMEMS. In common with other studies, see e.g. the review paper by Carrassi (2018), the method is illustrated using synthetic case when the true solution is known. The title uses words "model-to-model" to distinguish it from the usual "observations-to-model" DA. We think this is a fair and clear description of what this paper is about.

**Comment.** *What is the noise in the child model forecast meant to represent? In practice the child model will have errors of course, but it is unlikely they will be uncorrelated white noise.*

**Response.** The noise in the child model represents its own errors (as every model has) not the real features not resolved by the parent model. It is not just white noise. The errors in the child model include both uncorrelated (random Gaussian noise) and spatially correlated (bias, spatial shift) components in various combinations, see Figs 2,3 11 of the original MS. The last two are very important and well known sources of error in real models. As our results show, the white noise is a "worst case scenario".

**Comment.** *Line 250: this is the first mention of adding bias I think. This should be mentioned in the experimental set-up section*

**Response.** The bias is mentioned in line 192 of the original MS in the experimental set-up section. The text is made more clear as advised.

**Comment.** Line 256-258: *This statement seems strange. The field is constant away from the front so it wouldn't be hard to represent it.*

**Response.** We agree that the parent model represents well the field away from the front. The original MS says the same: Even at a resolution of 10 km that does not resolve the structure of the front (half-width of 6 km) the parent model gives a reasonable representation of areas outside the front where the changes in the state variable are smooth- see Fig. 1(b)

**Comment.** *Line 261: this stand-alone sentence ought to be attached to the subsequent paragraph.*

**Response .** Thank you, amended as advised.

**Comment.** *Figs 2 & 3: There's a lot of noise in the analysis. I would have thought if the error variances were specified correctly that this noise would be largely removed. How do you estimate the error variance of the S(y)?*

**Response.** The level of noise in the analysis is much reduced as compared to the child model output before the DA, which confirm that the SDDA method is reasonably formulated including the methods to compute matrices B and R. The algorithm to estimate the error variances in the parent model is explained in lines 150-153, 201-204 and 439-442 of the original MS and further clarified in the revised version.

**Comment.** *Fig 4(b): the caption doesn't mention the red line*

**Response.** Thank you. The caption is now corrected

**Comment.** Fig 5: You don't show the true field on the parent grid in this example which would be nice to see.

**Response.** The true field on the parent grid is now added to Fig5(d).

**Comment.** *Fig 6: why does the RMSE start rising again for eddy sizes > about 25 km?*

**Response.** The increase in the Assimilated RMSE to Non-Assimilated RMSE ratio for eddy sizes >25 km is very small. It is likely due to the increasing role of the correlated component of child model errors (spatial shift). In any case the improvement in RMSE after SDDA is greater than 10- fold across the whole range of sizes which shows good skill for SDDA.

**Comment.** *Line 366: "...random noise in the non assimilated model". Not sure what you mean there.*

**Response.** Thank you. This is a typo. We meant 'assimilated' rather than 'non-assimilated'. The text is corrected accordingly.

**Comment.** Fig 11: *The green line seems to indicate that the method can't deal very well with random noise.*

**Response.** When the errors in the child model are completely uncorrelated (random noise only) the improvement produced by SDDA is approximate 2-fold. This is the most difficult case for the SDDA. However, the actual child model is unlikely to have purely uncorrelated errors. The SDDA method is more efficient in a more realistic example of a combination of correlated ( spatial shift, bias) in uncorrelated (random noise) errors as shown in Figs. 2(d), 3(d), 4(b) , 6, 11.

**Comment.** *Generally the method seems to be dealing well with reducing the bias which is calculated over the whole domain so that positive/negative differences will average out.*

**Response.** The bias is actually calculated in a per-node basis. The reduction of bias is due to aggressive nudging of the bias in the child model to the bias of the parent model, which ( parent model) is assumed to be of good quality, see Eq (12).

**Comment.** Line 401: "the improvement is five to ten fold". Where do these numbers come from? They don't seem consistent with the ratio line in Fig. 12 (b).

**Response.** Fig 6 shows up to 5-fold ( even slightly better)  improvement in RMSE after SDDA **.** The phrase is corrected to state ' the improvement is up to five-fold'

**Comment.** Line 437: you should include references for these methods for error covariance estimation and also reference the review by Bannister (2008).

**Response .** The text is amended as advised.

**Comment.** *Line 438-439: I don't understand this sentence*

**Response .** The confusing sentence is removed as the following sentence conveys the message more clearly.

**Comment.** *Fig 14: The amplitude of the peak seems to be reduced in the SDDA method which should be mentioned*

**Response .** The text amended as requested.

**Comment.** *Line 467: It seems like this section comparing the SDDA method to the standard assimilation warrants a section of its own rather than being in the discussion section. The standard data assimilation chosen is only one of many available and the authors seem to have simplified its application significantly. Also, the SDDA method relies on the standard method producing a very good quality parent model solution.*

**Response.** The comparison of SDDA with the 'standard' method is a small amount of text and therefore fits well, in our opinion,  into the discussion section along with the comparison with the

spectral nudging method added in the revised MS as advised by the reviewer. The 'standard' method is selected as most widely used (Stewart et al, 2014; Carrassi et al 2018) and follows by a letter the textbook by Kalney (2003).

**Comment.** *Line 478: In most applications I know of, the model and observation error variances are not assumed to be homogeneous as is stated here. They would be estimated as spatially varying fields. If this was done, how would your conclusions be affected?*

**Response .** In his review paper, R.Bannister states that modern DA methods 'make assumptions about the way that background errors are spatially correlated (e.g. homogeneity and isotropy in the horizontal)' . In the SDDA method, the global homogeneity and isotropy are not assumed, but only the local one, in a small vicinity of each node, similar to the theory of turbulence, see e.g. a classical text by Tennekes and Lumley (Tennekes H., Lumley J.L, 1972. A First Course in Turbulence, MIT press, 300pp). To clarify, a new figure is added to the revised MS showing the spatial variations of the diagonal elements of B and R matrix.

**Comment.** Line 497: *Where were the observations located? Or did you assume observations to be available everywhere?*

**Response .** The 'observations' are located everywhere on the parent mesh. Clarification is added to the text.

**Comment.** Fig. 15(a). The quality of the figure is very poor and I found it hard to see the numbers or read any of the text. I couldn't tell where the observation error covariance was plotted as stated in the caption.

**Response.** The figure is updated and it is more clear in the revised MS. The authors will provide the production quality figures when and if the paper is accepted according to the journal publication policy.

**Comment.** *Eq on line 525-526: K is usually used as the notation for the Kalman gain matrix.*

**Response.** The letter K is often (but not always) used to denote the Kalman gain matrix. The MS uses the same notation (W) as in the referenced textbook by Kalney to avoid confusion.

**Comment.** *Fig 16: Could you show also the error (compared to the truth) in the standard method and the SDDA method?*

**Response.** The errors compared to truth are presented in Table 1 which seems to be adequate for the purpose.

**Comment.** *Line 542: It is stated that SDDA is more computationally efficient. Wouldn't it be fairer to compare the standard method with the total cost of assimilating the data into the parent model and then doing the SDDA?*

**Response.** The SDDA method is about assimilation of available data from the parent model into a high-resolution child model. The difficulty and cost of running and assimilating data into the parent model is beyond the scope of this paper. The SDDA method is intended for small groups that will not run the parent model. For them, the parent model output is the input data.